# Integrated analysis of cervical squamous cell carcinoma cohorts from three continents reveals conserved subtypes of prognostic significance

Human papillomavirus (HPV)-associated cervical cancer is a leading cause of cancer deaths in women. Here we present an integrated multi-omic analysis of 643 cervical squamous cell carcinomas (CSCC, the most common histological variant of cervical cancer), representing patient populations from the USA, Europe and Sub-Saharan Africa and identify two CSCC subtypes (C1 and C2) with differing prognosis. C1 and C2 tumours can be driven by either of the two most common HPV types in cervical cancer (16 and 18) and while HPV16 and HPV18 are overrepresented among C1 and C2 tumours respectively, the prognostic difference between groups is not due to HPV type. C2 tumours, which comprise approximately 20% of CSCCs across these cohorts, display distinct genomic alterations, including loss or mutation of the *STK11* tumour suppressor gene, increased expression of several immune checkpoint genes and differences in the tumour immune microenvironment that may explain the shorter survival associated with this group. In conclusion, we identify two therapy-relevant CSCC subtypes that share the same defining characteristics across three geographically diverse cohorts.

Despite screening and the introduction of prophylactic human papillomavirus (HPV) vaccination in developed countries, cervical cancer continues to be one of the leading worldwide causes of cancer-related deaths in women[1]. Prognosis for patients with metastatic disease remains poor, thus new treatments and effective molecular markers for patient stratification are urgently required. Cervical cancer is caused by at least 14 high-risk human papillomaviruses (hrHPVs), with HPV16 and HPV18 together accounting for over 70% of cases worldwide, with some variation by region[1–4]. Cervical squamous cell carcinoma (CSCC) is the most common histological subtype of cervical cancer, accounting for approximately 60–70% of cases, again with some variation seen across different populations[2]. Adeno- and adenosquamous histology are both associated with poor prognosis[5–8], while the relationship, if any, between HPV type and cervical cancer prognosis remains unclear[9]. HPV type is also associated with histology;

HPV16 and HPV18 were reported in 59.3% and 13.2% of CSCC and in 36.3% and 36.8% of adenocarcinoma respectively worldwide, between 1990 and 2010[2]. Previous landmark studies described the genomic landscape of cervical cancer in different populations[10–13] and in some cases identified subtypes based on gene expression, DNA methylation and/or proteomic profiles[10,11]. The Cancer Genome Atlas (TCGA) network identified clusters based on RNA, micro-RNA, protein/phosphoprotein, DNA copy number alterations and DNA methylation patterns and combined data from multiple platforms to define integrated iClusters[10]. In their analysis, only clustering based on the expression levels and/or phosphorylation state of 192 proteins as measured by reverse-phase protein array (RPPA) was associated with outcome, with significantly shorter overall survival (OS) observed for a cluster of cervical cancers exhibiting increased expression of Yes-associated protein (YAP) and features associated with epithelial-to-mesenchymal

e-mail: k.chester@ucl.ac.uk; a.feber@ucl.ac.uk; t.r.fenton@soton.ac.uk

transition (EMT) and a reactive tumour stroma. Since TCGA's RPPA analysis was restricted to 155 tumours including SCCs, adeno- and adenosquamous carcinomas, we set out to test the hypothesis that with data from more samples, we could identify a set of transcriptional and epigenetic features associated with prognosis within CSCC and to establish whether it is also present in independent patient cohorts representing different geographical locations and ethnicities. To identify molecular subtypes and prognostic correlates, we identified a set of 643 CSCCs (all HPV-positive), for which clinico-pathological data and genome-wide DNA methylation profiles were either publicly available or generated in this study, and for which in most cases, matched gene expression and somatic mutation data were also available (Table 1). Here, we show that CSCC samples from TCGA's cohort can be classified into one of two subgroups (C1 or C2), using gene expression or DNA methylation profiles and that the key features of these subgroups are conserved across validation cohorts from different continents. We identify differences in the tumour immune microenvironment and genetic alterations between subgroups and use detailed disease-specific survival data from our European cohort to validate the prognostic significance of the C1 / C2 classification.

## Results

### Identification of two gene expression-based clusters in cervical squamous cell carcinoma

Molecular and clinical differences between cervical adeno/adenosquamous and CSCCs are well documented[14–16] and gene expression differences were apparent in multi-dimensional t-distributed stochastic neighbour embedding (TSNE) analysis based on the top 10% most variable genes of three previously published cervical cancer cohorts[10,11,13] with available RNA-seq data (Supplementary Fig. 1a–d). To examine molecular and clinical heterogeneity specifically within SCC we focused all subsequent analysis on a collection of

**Table 1 | Summary of clinicopathological characteristics for five cervical cancer cohorts**

| | | Discovery Cohort | Validation Cohorts | | | | |
|---|---|---|---|---|---|---|---|
| | | TCGA | Bergen | Innsbruck | Oslo | Uganda | Total |
| Cohort Numbers | | 236 | 37 | 28 | 248 | 94 | 643 |
| Stage[a] | I Unknown | 7 (3%) | 0 | 1 (3.6%) | 0 | 0 | 8 (1.2%) |
| | IA | 1 (0.4%) | 0 | 1 (3.6%) | 0 | 0 | 2 (0.3%) |
| | IB | 115 (48.7%) | 34 (91.9%) | 14 (50%) | 20 (8.1%) | 15 (16.1%) | 198 (30.8%) |
| | II | 54 (22.9%) | 2 (5.4%) | 6 (21.4%) | 162 (65.3%) | 45 (47.9%) | 270 (42%) |
| | III | 39 (16.5%) | 0 | 4 (14.3%) | 56 (22.6%) | 31 (33%) | 130 (20.2%) |
| | IV | 14 (5.9%) | 1 (2.7%) | 2 (7.1%) | 10 (4%) | 2 (2.1%) | 29 (4.5%) |
| | NA | 7 (3%) | 0 | 0 | 0 | 1 (1.1%) | 8 (1.2%) |
| Age | Median (Range) | 47 (20–88) | 42 (28–64) | 49 (29–91) | 54 (22–82) | 45 (26–82) | |
| HPV Type | 16 | 136 (57.6%) | 22 (59.5%) | 17 (60.7%) | 178 (71.8%) | 39 (41.5%) | 392 (60.9%) |
| | 18 | 26 (11%) | 2 (5.4%) | 6 (21.4%) | 30 (12.1%) | 14 (14.9%) | 78 (12.1%) |
| | 45 | 19 (8.1%) | 4 (10.8%) | 0 | 9 (3.6%) | 14 (14.9%) | 46 (7.2%) |
| | Other | 55 (23.3%) | 9 (24.3%) | 5 (17.9%) | 31 (12.5%) | 27 (28.7%) | 127 (19.8%) |
| HPV Clade | Alpha 7 | 57 (24.2%) | 6 (16.2%) | 6 (21.4%) | 40 (16.1%) | 32 (34%) | 141 (21.9%) |
| | Alpha 9 | 172 (72.9%) | 29 (78.4%) | 20 (71.4%) | 194 (78.2%) | 54 (57.4%) | 469 (72.9%) |
| | Other | 7 (3%) | 2 (5.4%) | 2 (7.1%) | 14 (%) | 8 (8.5%) | 33 (5.1%) |
| Treatment[b] | Surgery | NA | 18 (48.6%) | 5 (17.9%) | 0 | NA | 23 (7.3%) |
| | Surgery + Radiotherapy | NA | 14 (37.8%) | 17 (60.7%) | 0 | NA | 31 (9.9%) |
| | Surgery + Radiotherapy + Chemotherapy | NA | 5 (13.5%) | 6 (21.4%) | 0 | NA | 11 (3.5%) |
| | Radiotherapy | NA | 0 | 0 | 47 (19%) | NA | 47 (15%) |
| | Radiotherapy + Chemotherapy | NA | 0 | 0 | 201 (81%) | NA | 201 (64.2%) |
| Survival[c] | Median (Range) | 1.9 (0–17.5) | 8 (1.8–13.2) | 9.8 (0.1–23.2) | 4.2 (0.3–12.7) | 1.1 (0–2.4) | |
| Vital status | Alive | 181 (76.7%) | 32 (86.5%) | 16 (57.1%) | 187 (75.4%) | 42 (44.7%) | 458 (71.2%) |
| | Dead | 55 (23.3%) | 5 (13.5%) | 12 (42.9%) | 61 (24.6%) | 52 (55.3%) | 185 (28.8%) |
| Cluster Assignment | C1 | 175 (74.2%) | 32 (86.5%) | 24 (85.7%) | 198 (79.8%) | 69 (73.4%) | 498 (77.4%) |
| | C2 | 61 (25.8%) | 5 (13.5%) | 4 (14.3%) | 50 (20.2%) | 25 (26.6%) | 145 (22.6%) |
| HIV Status | Positive | NA | NA | NA | NA | 59 | 59 |
| | Negative | NA | NA | NA | NA | 35 | 35 |
| Available Data | RNA-seq | 236 | 37 | NA | NA | 94 | 367 |
| | Methylation | 236 | 37 | 28 | 248 | 94 | 643 |
| | Mutation | 236 | 37 | NA | NA | 94 | 367 |
| | RPPA | 137 | NA | NA | NA | NA | 137 |
| | Microarray (Illumina WG-6 v3) | NA | NA | NA | 137 | NA | 137 |
| | Microarray (Illumina HT-12 v4) | NA | NA | NA | 109 | NA | 109 |
| | Copy number (MeCpG-derived) | 236 | 37 | 28 | 248 | NA | 549 |

[a]Further breakdown of tumour stage is included in Supplementary Data 1 and Supplementary Data 6.
[b]Treatment breakdown for individual patients is included in Supplementary Data 6.
[c]Overall survival was recorded for TCGA and Ugandan cohorts, while disease-specific survival was recorded for Bergen, Innsbruck and Oslo cohorts.

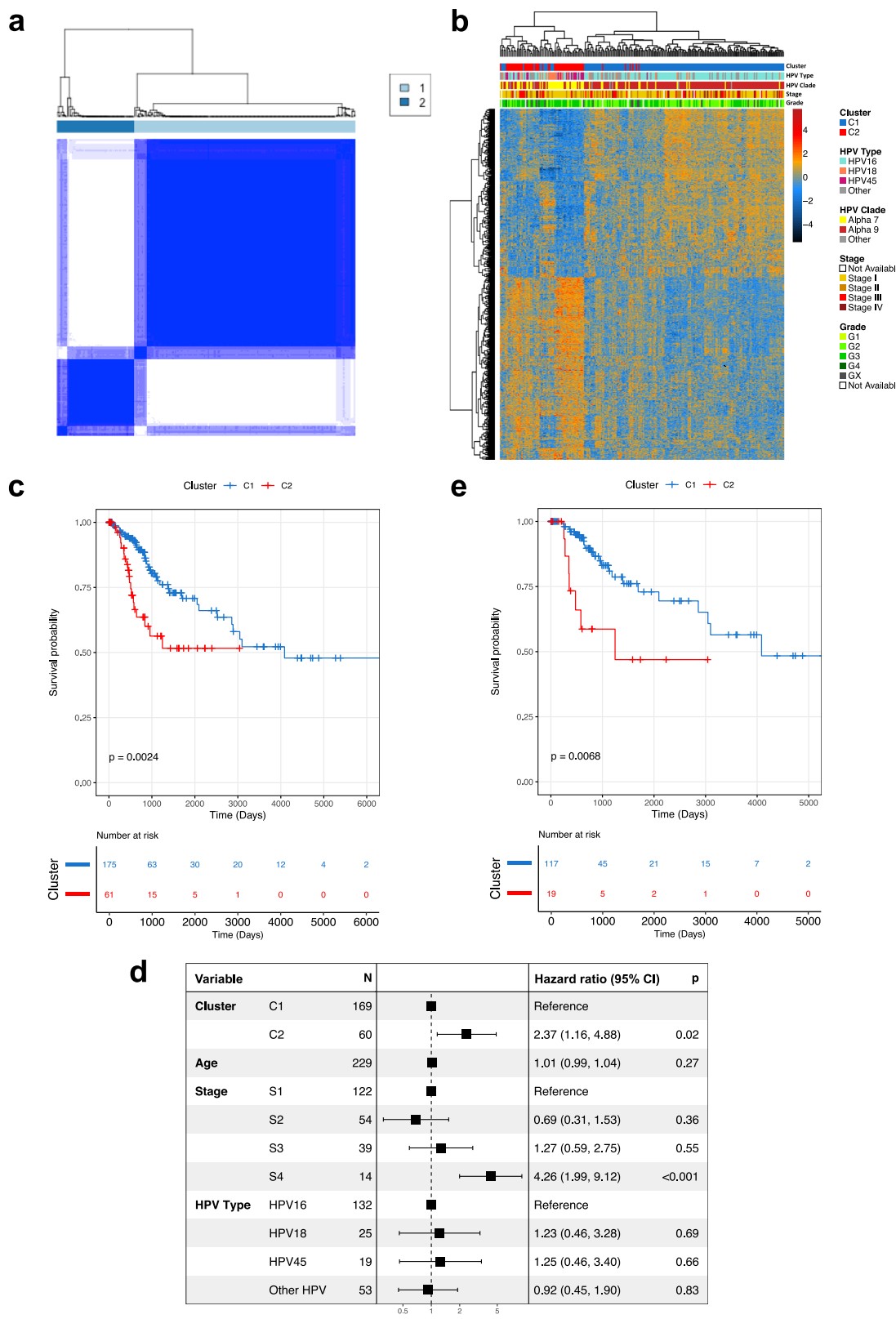

confirmed HPV-positive CSCCs from the USA, Europe and Uganda, as shown in Table 1.

236 cervical SCCs profiled by TCGA were defined as our discovery cohort (Table 1, Supplementary Data 1) and consensus clustering was performed using the top 10% most variable genes (*n* = 1377 genes, Supplementary Data 2). Consensus cluster membership heatmaps, delta area plot, consensus cumulative distribution function (CDF) and

proportion of ambiguous clusters (PAC) indicated the optimal number of clusters was two (Fig. 1a, Supplementary Fig. 2), the larger of which (*n* = 175) was designated C1 while the smaller cluster (*n* = 61) was designated C2 (Supplementary Data 1). Modelling transcriptomic differences between these two clusters identified 938 differentially expressed genes (DEGs, FDR = 0.01, FC > 2) (Fig. 1b, Supplementary Data 3). Tumours in C1 predominantly harboured HPV types from the

**Fig. 1 | Derivation of prognostic clusters in TCGA SCC cohort. a** Consensus clustering of 236 TCGA HPV + SCC patients produced an optimum 2 clusters (C1 and C2). Dark blue on the heat map represents patients (both row and column in heatmap) that clustered together 100% of the time while white represents patients that never clustered together after 90% sampling and replacement over 1000 iterations and considering only the 10% most variable genes. **b** There were 938 differentially expressed genes between the two clusters. Expression values for each of the 938 genes is represented by the rows in the heatmap (columns represent individual patients). **c** Kaplan–Meier curve shows that overall survival was higher in C1 patients in comparison to C2 patients when considering all patients. **d** C2 is an

individual predictor of prognosis, as is stage 4, while HPV type is not, as illustrated in a forest plot using cox-regression multivariate analysis using cluster type, age, stage and HPV type as covariates. Stage was not available for 7 patient and so they were excluded from the analysis reducing the cohort to 229. The box and whiskers for each represent the hazard ratio (HR) and 95% CI respectively. The x axis represents the HR in the log10 scale. **e** Survival analysis using a Kaplan–Meier curve shows that overall survival was higher in C1 patients when considering only HPV16 + tumour patients. *P* values on Kaplan–Meier curves from two-sided log-rank test, *p* values in forest plot from two-sided Wald test. Source data are provided as a Source Data file.

**Table 2 | Overall (TCGA) and disease-specific (individual and combined European cohorts) survival analyses**

| | | Univariate | | | Multivariate | | |
|---|---|---|---|---|---|---|---|
| | | HR | *p* value (Log rank) | 95% CI | HR | *p* value (Wald) | 95% CI |
| TCGA[a] | C1 | – | – | – | – | – | – |
| | C2 | 2.37 | 0.003 | 1.42, 4.56 | 2.37 | 0.019 | 1.16, 4.88 |
| Bergen[b] | C1 | – | – | – | – | – | – |
| | C2 | 5.28 | 0.07 | 0.87, 31.9 | 98.1 | <0.001 | 8.41, 1145 |
| Innsbruck[b] | C1 | – | – | – | – | – | – |
| | C2 | 0.55 | 0.6 | 0.07, 4.26 | 1.23 | 0.9 | 0.06, 24.9 |
| Oslo[b] | C1 | – | – | – | – | – | – |
| | C2 | 1.69 | 0.07 | 0.96, 3.14 | 2.17 | 0.019 | 1.14, 4.13 |
| Europe combined[b] | C1 | – | – | – | – | – | – |
| | C2 | 1.67 | 0.05 | 0.99, 2.90 | 2.32 | 0.004 | 1.30, 4.13 |

[a]Multivariate analysis included age, tumour stage and HPV type as covariates.
[b]Multivariate analysis included age, tumour stage, HPV type and treatment regimen as covariates.

HPV16-containing alpha-9 clade (150/175) while 38 of 61 C2 tumours contained HPV types from the HPV18-containing alpha-7 clade. C2 tumours were 13.3 times more likely to harbour alpha 7 HPVs than C1 tumours ($p = 1.8 \times 10^{-14}$, Fisher's Exact Test) (Fig. 1b).

Univariate analysis of overall survival (OS) revealed worse outcomes for patients with C2 tumours (Hazard ratio (HR) = 2.37, log-rank test $p = 0.002$, 95% CI 1.42, 4.56; Fig. 1c) and in Cox regression including age, tumour stage and HPV type as covariates along with cluster membership ($n = 229$ patients for whom data on all covariates were complete), only membership of the C2 cluster (HR = 2.37, Wald test $p = 0.019$, 95% CI 1.16, 4.88) and a tumour stage of IV (HR versus stage I = 4.26, Wald test $p < 0.001$, 95% CI 1.99, 9.12) were independent predictors of OS (Table 2, Fig. 1d). The 3-year survival rate of the C1 subgroup was 79% compared to 56% for the C2 subgroup, while the 5-year survival rate was 71% and 52% for C1 and C2 subgroups respectively. The relationship between cluster and OS is also clear when restricting the analysis to HPV16-containing tumours in each cluster in both univariate analysis (HR = 3.13, log-rank test $p = 0.007$, 95% CI 1.31, 7.46; Fig. 1e) and multivariate analysis, including age and tumour stage as covariates (HR = 3.76, Wald test $p = 0.004$, 95% CI 1.54, 9.19; Supplementary Data 4).

**Identification of C1 and C2 CSCCs and association with prognosis in independent SCC cohorts**

To further investigate the association between C1/C2 cluster membership and OS, we assembled a combined validation cohort consisting of 313 CSCC patients treated at three centres in Europe (Bergen ($n = 37$), Oslo ($n = 248$) and Innsbruck ($n = 28$)), for which much more detailed clinical information, including disease-specific survival (DSS), was available than in TCGA and for which genome-wide DNA methylation profiles from Illumina Infinium 450k arrays (the same platform used by TCGA) were either available or generated in this study (Table 1). Since RNA-seq data were not available for all European samples, cluster membership was assigned using a support vector

machine (SVM) classification model based on 129 CpG sites (methylation variable positions, MVPs) at which methylation differed significantly between tumours in C1 vs C2 clusters in the discovery cohort (Fig. 2a, b; mean delta-Beta > 0.25, FDR < 0.01, Supplementary Data 5), 18 of which were located within 12 genes differentially expressed between the clusters (Supplementary Fig. 3). MVP and DEG signatures were also used to assign cluster membership to 94 CSCCs from a Ugandan cohort originally profiled by the Cancer Genome Characterization Initiative (CGCI)[11], for which both DNA methylation and RNA-seq data were available. C2 tumours from all cohorts clustered together using TSNE analysis based on the MVP signature (Fig. 2c) and high concordance between DEG and MVP-based cluster allocation was observed in all cohorts for which both gene expression (RNA-seq for Uganda and Bergen or Illumina bead chip arrays for Oslo) and DNA methylation data were available (Supplementary Fig. 4a, b). Single-sample gene set enrichment analysis (ssGSEA) confirmed differential expression of the signature genes in tumours classified as C1 or C2 using DNA methylation data (Supplementary Fig. 4c). 59 of 313 (18.8%) tumours in the combined European cohort (Fig. 2b, Supplementary Data 6) and 25 of 94 (26.6%) tumours in the Ugandan cohort were classified as C2 (Table 1, Supplementary Fig. 5a, Supplementary Data 6). As in the discovery cohort, most C1 tumours from the European and Ugandan cohorts harboured alpha-9 HPV types (260/325) while C2 tumours were 3.9 times more likely to harbour alpha-7 HPVs than C1 tumours ($p = 1.07 \times 10^{-6}$, Fisher's Exact Test) (Fig. 2b, Supplementary Fig. 5a). Interestingly 80% (20/25) of Ugandan C2 patients were human immunodeficiency virus (HIV) positive, while only 56% (39/69) of C1 patients were HIV positive (OR = 0.33; $p = 0.05$, Fisher's Exact Test, Supplementary Fig. 5a).

Univariate analysis indicated lower DSS in C2 tumours from the European cohort (HR = 1.67, log-rank test $p = 0.05$, 95% CI 0.99, 2.90; Fig. 2d) and Cox regression controlling for FIGO stage, age, HPV type and treatment (surgery alone, surgery with radio-chemotherapy, surgery with radiotherapy alone, radio-chemotherapy and radiotherapy

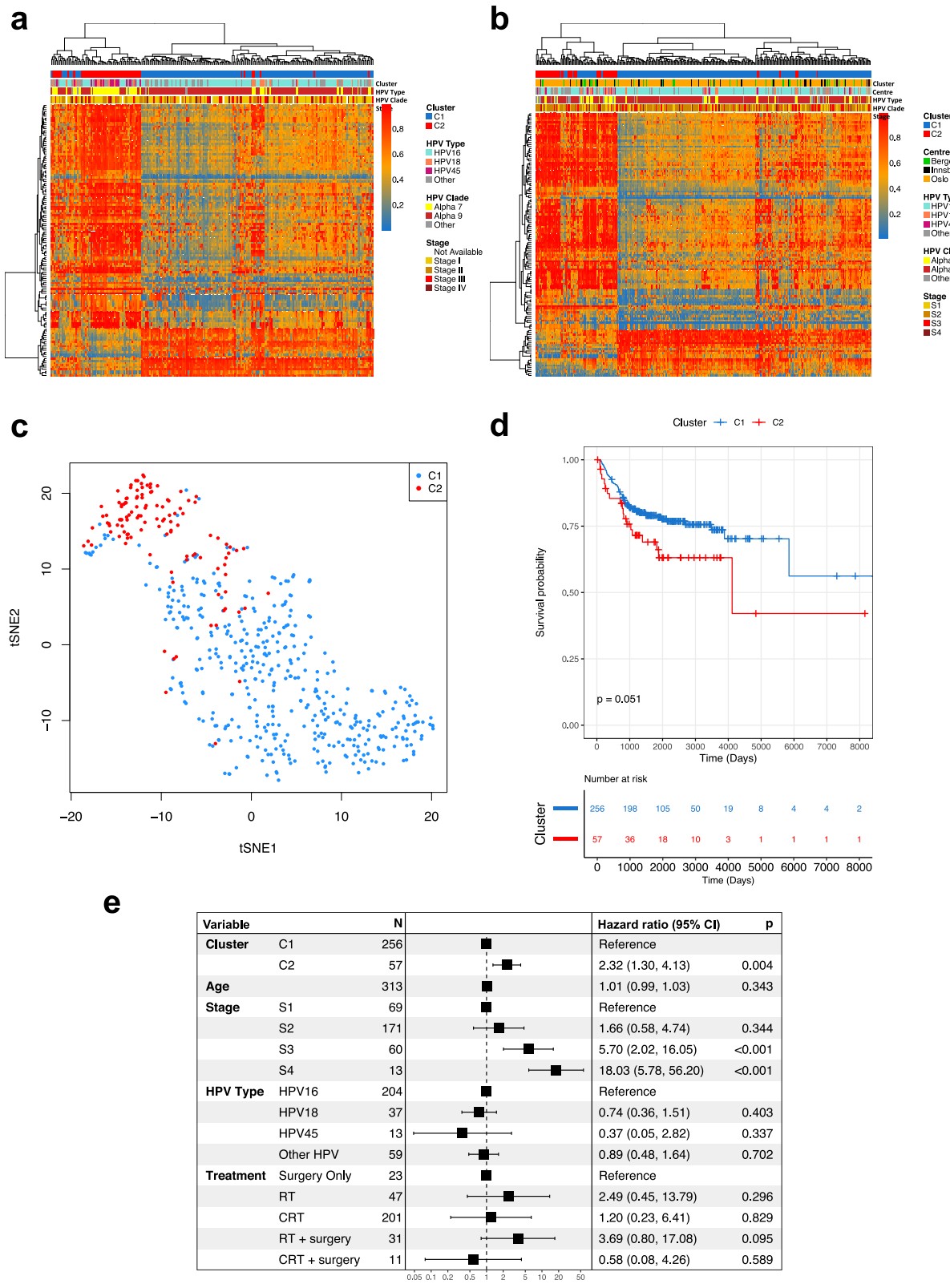

alone) identified C2 status but not HPV type to be an independent predictor of DSS (HR = 2.32, Wald test *p* = 0.004, 95% CI 1.3, 4.13) along with the expected predictor, tumour stage (Fig. 2e, Table 2). The 3-year disease-specific survival rate was 81% for C1 patients compared to 72% for C2 patients while 5-year disease-specific survival was 79% and 66% for C1 and C2 subgroups respectively. As in the discovery cohort, a significant prognostic difference was identified between the C1 and C2 subgroups when considering only the HPV16-positive tumours (*n* = 204) (univariate HR = 2.15, log-rank test *p* = 0.036, 95% CI 1.04, 4.41; multivariate HR = 2.04, Wald test *p* = 0.084, 95% CI 0.91, 4.6; Supplementary Data 4, Supplementary Fig. 5b). Interestingly the prognostic difference was even greater among 78 patients in the European cohort that did not receive chemotherapy (Supplementary Fig. 5c; univariate HR = 3.01, log-rank test *p* = 0.01, 95% CI 1.29, 7.01;

**Fig. 2 | Cluster allocation of validation cohorts using methylation signature.** A DNA methylation based signature of 129 methylation variable positions (MVPs; dB > 0.25, FDR < 0.01), represented by rows in heatmaps, separates clusters C1 and C2 patients in the TCGA cohort ($n = 236$) (**a**) and the validation dataset from the 3 European centres ($n = 313$) (**b**). **c** Using DNA methylation data (BMIQ corrected beta values) for the TCGA and European cohorts, C2 tumours cluster together based on the 129 MVP signature. **d** The disease-specific survival (DSS) for C1 patients is higher than for C2 patients in the combined European validation cohort. **e** C2 is an

individual predictor of prognosis, as is stage 3 and 4, while HPV type is not, as illustrated in a forest plot using cox-regression multivariate analysis using cluster type, age, stage, HPV type and treatment regimen (RT = radiotherapy, CRT = chemoradiotherapy) as covariates. The box and whiskers for each represent the hazard ratio (HR) and 95% CI respectively. The x axis represents the HR in the log10 scale. P values on Kaplan–Meier curves from log-rank test, p values in forest plot from Wald test. Source data are provided as a Source Data file.

multivariate HR = 2.94, Wald test $p = 0.032$, 95% CI 1.10, 7.86). The low overall survival rate among patients in the CGCI (Ugandan) cohort relative to the other cohorts (Supplementary Fig. 5d), the lack of disease-specific outcome data and the fact that 20 of the 25 patients with C2 tumours were also HIV+ precluded us from analysing the relationship between cluster and prognosis in this cohort. Taken together, the C1/C2 clusters identified in the TCGA cohort (USA) are apparent in cohorts of CSCC patients from Europe and Uganda and tumours can be accurately assigned to cluster using either gene expression or DNA methylation profiles. Unlike HPV type, C1/C2 cluster is an independent predictor of OS in the TCGA discovery cohort ($n = 236$) and, more importantly, of DSS in the European validation ($n = 313$) cohort. Consistent with this, there is no difference in the breakdown of C1 and C2 tumours by stage in any of the cohorts and although higher patient numbers are required to permit robust stage-by-stage survival analysis, a clear trend of higher survival rates for C1 patients is observed within each tumour stage (Supplementary Data 7). Note also that the good prognosis associated with cluster C1 is not simply attributable to an enrichment among small tumours that are curable by surgical resection alone, as only 2 of the 200 stage I cases for which detailed staging was available were stage IA (Table 1).

## Relationships between C1/C2 and clusters previously identified by TCGA

Of the 178 tumour samples that made up the core set in the TCGA's landmark study into cervical cancer genomics/epigenomics[10], 140 CSCCs were present in our discovery cohort of 236 (Supplementary Data 8). This enabled comparisons between our gene expression-based cluster allocations and the subtypes defined by TCGA (Fig. 3). TCGA analysis included integrated clustering using multi-omics data (three iClusters, two of which ('keratin-high' and 'keratin-low') were composed entirely of CSCCs) and clustering based on transcriptomic data (three mRNA clusters). There is considerable overlap between our C1 cluster and TCGA's 'mRNA-C2' cluster (84/106, odds ratio (OR) = 9, $p = 3.1 \times 10^{-7}$, Fisher's Exact Test) and keratin-high iCluster (80/106, OR = 44.7, $p = 9.8 \times 10^{-13}$, Fisher's Exact Test), and between our C2 cluster and TCGA's 'mRNA-C3' cluster (19/34, OR = 8.15, $p = 1.7 \times 10^{-6}$, Fisher's Exact Test) and keratin-low iCluster (27/34, OR = 18.3, $p = 5.02 \times 10^{-11}$, Fisher's Exact Test). Neither the 'mRNA-C3' nor the keratin-low iCluster were associated with poor prognosis in TCGA's analysis and given the increased expression of a subset of keratin genes (including *KRT7*, *KRT8* and *KRT18*) in TCGA tumours C2 according to our classification (Fig. 3), we decided against adopting the keratin-high / keratin-low nomenclature for our clusters. We also examined the relationship between our subtypes and three clusters defined by TCGA based on reverse phase protein array (RPPA) data. Notably, 57% of TCGA tumours that we classified as C2 with RPPA data available belong to the TCGA 'EMT' cluster compared with only 25% of tumours that we classified as C1 (Fig. 3). Consistent with the proteomic classification, when considering all 140 TCGA SCCs for which TCGA reported RNA-seq-based EMT scores[10], those in our C2 group display higher scores than those we classified as C1 (Supplementary Fig. 6). Although there is greater concordance between our C2 group and the TCGA EMT cluster compared to C1, it is clearly distinct from TCGA's EMT cluster.

## Genomic analyses of prognostic clusters

To investigate whether C1 and C2 tumours differ at the genomic level in addition to the transcriptomic and epigenomic differences observed above, whole-exome data was obtained for SCCs from three cohorts, TCGA[10], Bergen[13] and Uganda[11]. This amounted to 367 samples, 29 of which were classed as hypermutated by standards set by TCGA[10] (>600 mutations). The median tumour mutation burden (TMB) was 2.04/Mb for all tumour, 2.11/Mb for C1 tumours and 1.82/Mb for C2 tumours (1.92/Mb, 1.94/Mb and 1.72/Mb respectively after removal of hypermutated samples). We detected four mutation signatures for the combined cohorts (Supplementary Fig. 7): as expected based on previous studies[10,11,13], COSMIC signatures 2 and 13 (characterised by C > T transitions or C > G transversions respectively at TpC sites attributed to cytosine deamination by APOBEC enzymes); age-related COSMIC signature 1 (characterised by C > T transitions attributed to spontaneous deamination of 5' methylated cytosine) and COSMIC signature 5, for which the underlying mutational process is unknown[17] (https://cancer.sanger.ac.uk/signatures/). The proportion of mutations attributable to each signature did not vary between clusters (Fig. 4).

Having excluded the hypermutated samples, we next performed dNdScv analysis[18] on each cohort, followed by p value combination using sample size weighted Fisher's method followed by FDR correction[19] to permit identification of significantly mutated genes (SMGs) across the entire dataset. This combined approach, followed by analysis of individual samples by cluster identified 34 SMGs (Fig. 4, Supplementary Data 9), 21 of which (highlighted by †) have not previously been identified as SMGs in cervical cancer[10,11,13]. Of the 34 SMGs, 21 listed in blue in Fig. 4 were significantly mutated in C1 tumours, two genes (*STK11* and *NF2*, listed in red) were SMGs only in C2 and a further three genes (*FAT1*, *HLA-A* and *FRG1*) were significantly mutated in the individual analyses for both C1 and C2. Eight genes were significantly mutated only when both C1 and C2 clusters were combined in a single analysis (Fig. 4, Supplementary Data 9), while a further two genes (*MAGEB6* and *DNAJB1*) were categorised as SMGs in the C1 cluster but were no longer significant when C2 tumours were included in the analysis. The frequency of mutations in SMGs that had been previously observed was comparable between our combined cohort and each respective SMG study (Supplementary Data 10). Among the 21 genes that have not previously been identified as significantly mutated in cervical cancer, six are SMGs in other SCCs, including head and neck (*NOTCH1*, *JUB* (also known as *AJUBA*), *MLL2* (also known as *KMT2D*), *RB1*, *PIK3R1*)[20], oesophageal (*MLL2*, *NOTCH1*, *RB1*)[21] and lung SCC (*NOTCH1*, *RB1*, *MLL2*, *CREBBP* (also known as *KAT3A*))[22]. Conversely, several genes previously identified as SMGs in cervical cancer but not in our analysis, including *TP53*, *ARID1A* and *TGFBR2* are significantly mutated in adenocarcinoma but not in CSCC[10,13]. Comparing somatic mutation rates in SMGs between clusters using binomial regression identified *PIK3CA* (FDR = 0.001) and *EP300* (FDR = 0.046) mutations as disproportionally more common in C1 tumours and *STK11* (FDR = 0.005) and *NF2* (FDR = 0.045) as enriched in C2 tumours (Fig. 4). *STK11* is also under-expressed in C2 tumours compared with C1 tumours (Supplementary Data 3).

## C2 tumours display Hippo pathway alterations and increased YAP1 activity

Two SMGs from our analysis (*LATS1* and *NF2*) are core members of the HIPPO signalling pathway, while SMGs *FAT1*, *JUB* and *STK11* are known

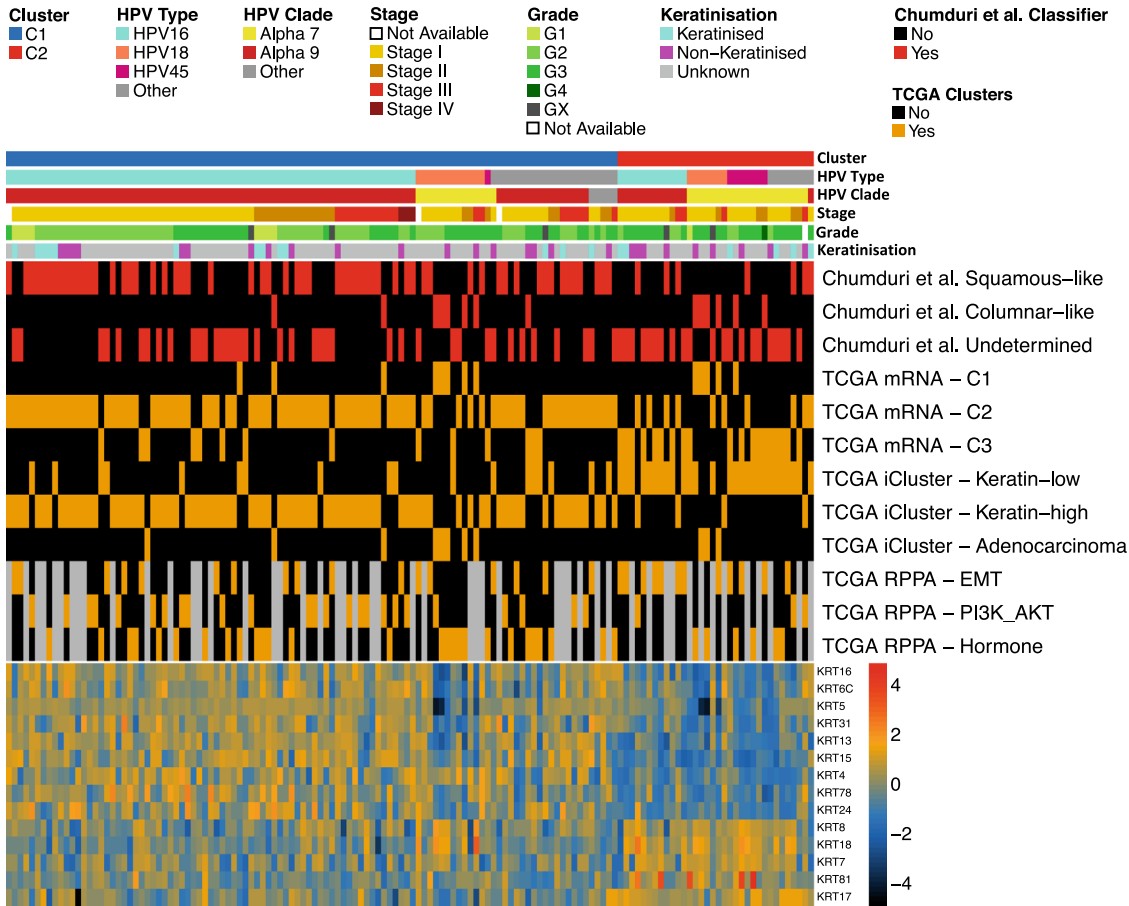

**Fig. 3 | Comparison of SCC subgroups with previous studies.** Cluster analysis had previously been performed on 140 TCGA SCC tumours in two studies – one determined clusters based on cell of origin markers (Chumduri et al., 2021, red), and one determined clusters based on integrated omics data (TCGA Network, 2017, orange). Patients (columns) are identified as belonging to one of the clusters identified by Chumduri *et al* by red in the appropriate row and belonging to one of the TCGA clusters by orange in the appropriate row. C2 patients were more likely to belong to the Chumduri *et al* columnar-like cluster than the squamous-like cluster (odds ratio (OR) = 9.95, *p* = 0.006, Fisher's Exact Test) and more likely to belong to the TCGA keratin-low iCluster (OR = 18.3, *p* = 5.02 ×10[−11], Fisher's Exact Test) than other iClusters. The heatmap at the bottom of plot represents expression levels of cytokeratin genes present in our C2 gene signature. 9 of the 14 genes exhibit low expression in C2 tumours, and 5 are highly expressed and so the TCGA "keratin-low" nomenclature was not used despite the large overlap with our C2 cluster. All statistical tests were two-sided. Source data are provided as a Source Data file.

regulators of HIPPO signalling[23–25]. Mutations in *LATS1*, *FAT1*, *JUB*, *STK11* or *NF2* (the latter two of which are significantly mutated specifically in C2 tumours, Fig. 4) result in aberrant activation of the downstream transcription factor, Yes1 associated transcriptional regulator (YAP1)[26–30], the expression of which is also elevated at the mRNA level in C2 tumours (Supplementary Data 3).

We generated segmented copy number data for all tumours (combining TCGA and European validation cohort samples for which the necessary data were available for maximum statistical power), which identified 211 focal candidate copy number alterations (CNAs) at FDR < 0.1. Following binomial regression, we identified five discrete CNAs that differed in frequency between C1 and C2 clusters (Fig. 5a; FDR < 0.1, log2 (Odds Ratio) >1). All five were more prevalent in C2 tumours and included 11q11 and 1q21.2 deletions and 6p22.1, 11q22.1 and 11q22.2 gains. 11q22.2 contains matrix metalloproteinase genes (MMPs) which are well known to be involved in metastasis[31], but notably 11q22.1 contains the *YAP1* gene. Furthermore, analysis of Reverse Phase Protein Assay (RPPA) data from TCGA revealed significantly higher YAP1 protein expression in C2 tumours (Fig. 5b). We confirmed that of the 137 TCGA cases for which RPPA data were available, cases with *YAP1* amplification (8/37 C2 tumours and 6/100 C1 tumours) also showed increased YAP1 mRNA and protein expression (Supplementary Fig. 8). In total 10 genes from a 22 gene signature that

predicts HIPPO pathway activity in cancer[32] are differentially expressed between C1 and C2 tumours (Supplementary Data 3).

## Differences in the tumour immune microenvironment between C1 and C2 tumours

The presence of circulating HPV-reactive T-lymphocytes and of tumour-infiltrating cytotoxic T-lymphocytes have been associated with lower N-stage and improved prognosis in cervical cancer patients[33–41]. We used DNA methylation data to compare the cellular composition of TCGA tumours[42], observing differences in the proportions of multiple cell types between the subgroups (Fig. 6a); most notably decreased CD8 + (cytotoxic T lymphocytes (CTL)), and a marked elevation of neutrophil and CD56 + natural killer (NK)-cells in C2 tumours. Repeating this method with the validation cohorts produced results that were remarkably similar (Fig. 6b). Differences in the proportions of cell types between C1 and C2 in the validation cohort mirrored those in the TCGA cohort, decreased CTL, and elevated neutrophil, NK-cell and endothelial cell levels were observed in C2 tumours. Importantly, this was not driven by any single validation cohort, as individual cohorts displayed consistent patterns of differences in the proportion of cell types between C1 and C2 tumours, especially with regards to CTLs, neutrophils and NK-cells (Supplementary Fig. 9a–d). C2 tumours also exhibit markedly higher

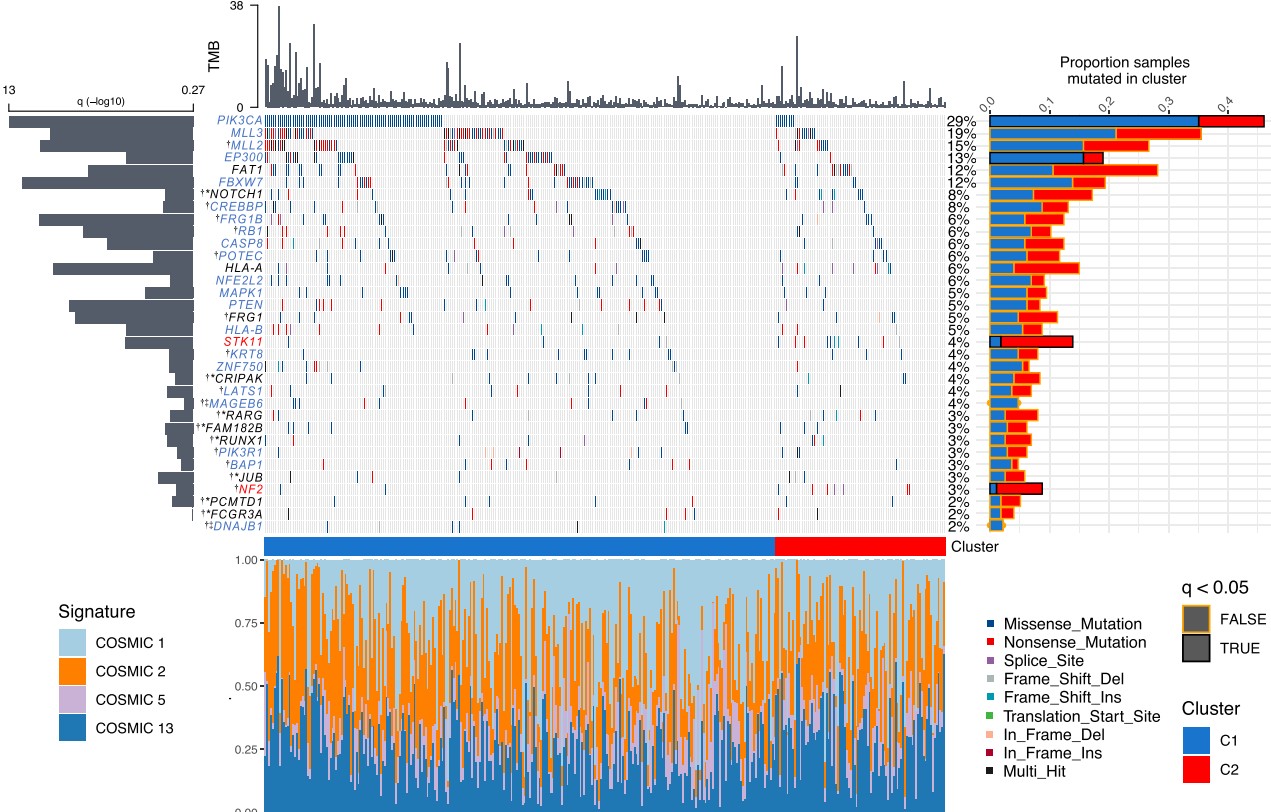

**Fig. 4 | Genomic summary of significantly mutated genes (SMGs) in SCC cohorts.** The central plot shows mutation type and frequencies for 34 SMGs identified using dNdSCV on TCGA, Bergen and Ugandan cohorts (367 total patients). The histogram above the main plot represents tumour mutation burden (TMB, mutations/megabase) per sample. The histogram to the left displays the statistical significance of each SMG while the histogram to the right represents the proportion of C1 (blue) and C2 (red) tumours in which each gene is mutated. Black outlines around bars in this histogram signify a significant difference (FDR < 0.05, Wald Test) in mutation frequency between clusters. The stacked histogram below the main plot represents the breakdown of single base mutational signatures (SBS) detected in each tumour. Gene name key: blue = SMG in C1 tumours; red = SMG in C2 tumours; black = SMG in C1 and C2 tumours; black * = SMG only when combining C1 and C2 tumours; † = SMG not previously reported in cervical cancer; ‡ = SMG in C1 tumours but not significant when C2 tumours also included.

neutrophil:CTL ratios (Supplementary Fig. 10a, b) and neutrophil:lymphocyte (CTL, B-cell and Treg) ratios (NLR, Supplementary Fig. 10c, d); established adverse prognostic factors in cervical cancer[43–45]. At 0.7, the NLR in C1 tumours across all cohorts was less than half that observed in C2 tumours (1.85). Differences in the abundance of infiltrating immune cells between C1 and C2 were maintained upon restricting the analysis to HPV16-containing tumours (Supplementary Fig. 9e, f), thus eliminating the possibility that they were were due to the overrepresentation of alpha-9 HPV types in C1 and alpha−7 in C2 tumours.

Validation of MethylCIBERSORT cell estimates was performed for a subset of samples from the Innsbruck cohort using CD8 (CTLs) and myeloperoxidase (MPO, neutrophils) immunohistochemistry (IHC)-based scores from a pathologist blinded to cluster designation (Supplementary Fig. 11a–c) and for CTLs in the Oslo cohort samples using comparison of MethylCIBERSORT estimates to CD8 IHC-based digital pathology scores (Supplementary Fig. 11d).

Also of potential significance regarding the tumour immune microenvironment, is the presence of two immune checkpoint genes, CD276 (also known as B7-H3) and NT5E (also known as CD73) in the set of 938 signature DEGs that separate the clusters (Supplementary Data 3). Both B7-H3 and NTSE, along with a third immune checkpoint gene (PD-L2) are expressed at higher levels in C2 tumours (Supplementary Fig. 12) and hypomethylation of two CpGs in the NT5E promoter is evident in C2 tumours (Supplementary Data 5). All three suppress T-cell activity[46–48] and B7-H3 expression has been linked to poor prognosis in cervical cancer[49,50].

## Evidence for differences in stromal fibroblast phenotype between C1 and C2 tumours

Gene set enrichment analysis using Metascape[51] suggested increased EMT (Supplementary Data 11) in C2 tumours, with 52 of 200 genes in in the EMT Hallmark gene set upregulated. As noted above there is also greater overlap between the C2 cluster and an EMT cluster defined by TCGA and based on RPPA data (Fig. 3). Single-cell RNA sequencing and xenografting studies strongly suggest that rather than arising from the tumour cells (few of which have undergone EMT at any given time[52–54]), mesenchymal gene signatures in bulk tumour expression data instead derive from stromal fibroblasts, which can adopt various phenotypes and play an important role in shaping the tumour immune microenvironment[55,56]. In addition to YAP1, which has been linked to the formation of cancer-associated fibroblasts (CAFs)[57] (as well as EMT[58–60] and angiogenesis[61]), C2 tumours display increased expression of the CAF marker genes FAP and SERPINE1 (also known as PAI-1)[62]; the latter evidenced at both mRNA and protein levels (Supplementary Data 3, Fig. 5b). Overall fibroblast content as estimated by MethylCIBERSORT is similar between C1 and C2 tumours (Fig. 6a, b) but given recent findings regarding the extent and prognostic significance of CAF heterogeneity in the tumour microenvironment[63–67], we hypothesised that CAF phenotype rather than overall abundance, may differ between C1 and C2 tumours. To examine this, hierarchical clustering was performed based on the expression of eight gene sets (68 genes) curated by Qian et al.[64], representing CAF-related biological processes and which are differentially expressed across six CAF phenotypes recently identified in a pan-cancer analysis[67]. C2 tumours clustered

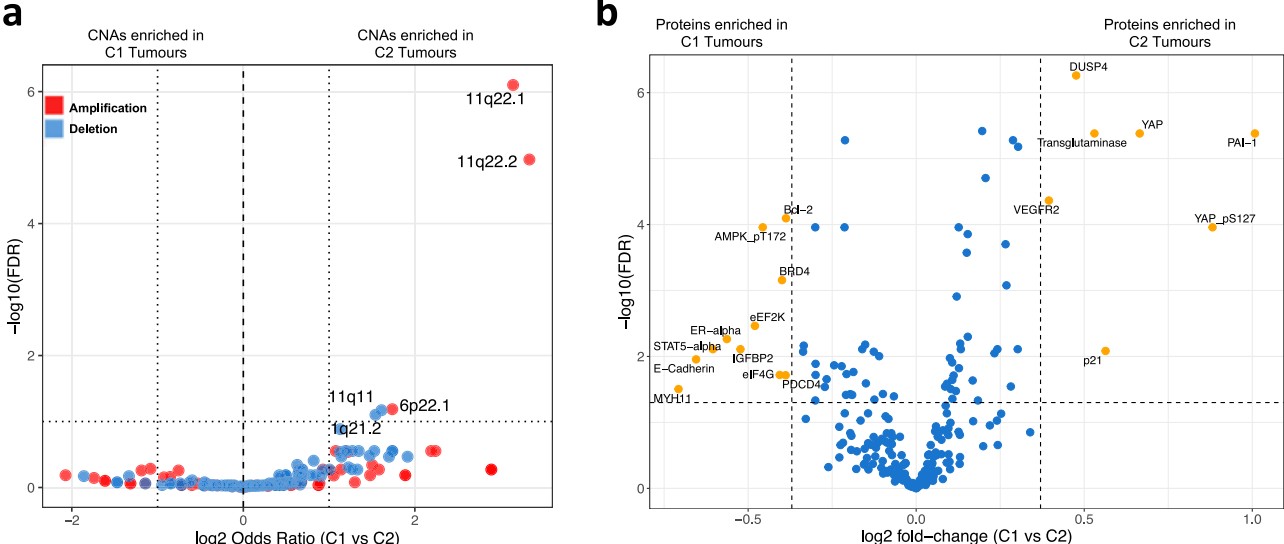

**Fig. 5 | Copy number and protein level differences between SCC subgroups. a** 3 copy number amplifications (red points) and 2 copy number deletions (blue points) were significantly enriched (Wald Test, FDR < 0.05, horizontal dotted line) in C2 tumours as determined by binomial regression. GISTIC copy number peak frequencies between C1 and C2 tumours, also had to have a log2 odds ratio (x axis) >1 or <−1 (vertical dotted lines) to be considered as enriched in either cluster. **b** Using reverse phase protein array data for 140 CSCCs in the TCGA cohort 18 proteins are differentially expressed between C1 and C2, 7 proteins upregulated in C2 and 11 upregulated in C1 tumours. Orange points represent proteins that are expressed at a fold change of >1.3 or a fold change < 1.3 and where the FDR is <0.05 (*t*-test). Source data are provided as a Source Data file.

together, displaying increased expression of proinflammatory genes associated with an inflammatory (pan-iCAF2) CAF phenotype, while contractile genes associated with a myofibroblast CAF phenotype (pan-myCAF) were more highly expressed in C1 tumours (Fig. 6c). Myosin heavy chain 11 (*MYH11*), identified as a marker of the myofibroblast CAF phenotype in multiple tumour types[63,67,68] displays elevated protein expression in C1 tumours (Fig. 5b). C1 tumours also display increased expression of collagens and other ECM genes associated with the pan-dCAF phenotype and of proliferative genes linked to the pan-pCAF phenotype[67] (Fig. 6c). Single cell RNA sequencing of head and neck squamous cell carcinoma showed that stromal myofibroblasts are transcriptionally distinct from other CAFs[55] and consistent with this, C2 tumours are 4.8× (*p* = 1.78 × 10⁻⁹, Fisher's Exact Test) more likely to be classified as 'CAF-high' than C1 tumours using a four-gene CAF index defined by Ko et al.[56]. Indeed, three of the four CAF index genes (*TGFBI, TGFB2* and *FN1*) appear in the 938 DEG signature that separates C2 from C1 tumours (Supplementary Data 3).

## Discussion

In this study we hypothesised that by drawing upon several cervical cancer cohorts for which -omics data, clinical information and HPV typing were either available or for which we were able to profile samples ourselves, we would be able to gain further insight into CSCC – the most common histological cervical cancer subtype. Clustering of CSCCs according to the 10% most variable genes identified two clusters (C1 and C2) that bear resemblance to the keratin-high and keratin-low iClusters originally defined by TCGA[10]. Cluster membership is an independent predictor of OS and CSCCs can be accurately assigned to cluster using either a 938 gene expression signature or a 129 MVP DNA methylation signature, providing a means by which to gain prognostic information for cervical cancer patients. While HPV16 and the alpha-9 clade to which it belongs have been associated with longer PFS and OS in several studies[69–74], the relationship between HPV genotype and cervical cancer prognosis remains unclear, as highlighted by a recent meta-analysis[9]. In our multivariate analyses, membership of the C2 cluster but not HPV type was an independent predictor of poor prognosis in both the discovery and validation cohorts and remained so when only HPV16-positive tumours in either cohort were

considered. Possibly, the reason that HPV16 and other alpha-9 HPV types have been associated with more favourable outcomes is that these viruses are more likely to cause C1-type tumours. Although larger numbers are needed for robust within-stage comparisons of C1 and C2 tumours, we observe a clear trend in the survival rates between C1 and C2 by stage (Supplementary Data 7). Taking molecular (C1/C2) subtyping into account may therefore allow for more accurate prognostication than current staging and potentially (clearly dependent upon prospective studies) different clinical management of patients with C1 versus C2 tumours. This could include the identification of patients at risk of relapse and who may therefore require further adjuvant therapy after completion of upfront therapy.

Adeno- and adenosquamous carcinomas, which are thought to arise from the columnar epithelium of the endocervix, have been linked to poor prognosis in cervical cancer[5–8] and to avoid differences due to histology, we focused our study entirely on CSCC. Interestingly, of the 14 keratin genes that are differentially expressed between C1 and C2 tumours, three (*KRT7, KRT8* and *KRT18*) that are upregulated in C2 were classified as marker genes for columnar-like tumours with a possible endocervical origin in a recent study that used single cell RNA-sequencing and lineage tracing experiments to explore cell-of-origin for CSCC and adenocarcinoma[75]. In contrast, C1 tumours display increased expression of *KRT5*, a marker of the squamous-like subtype with a proposed ectocervical origin identified by Chumduri et al.[75] (Fig. 3). Other signature genes (*TP63, CERS3, CSTA, CLCA2, DSC3* and *DSG3*) upregulated in C1 tumours are also markers of the squamous-like subtype, while further columnar-like marker genes (*MUC5B* and *RGL3*) are upregulated in C2 tumours (Supplementary Data 3). Squamous-like tumours are significantly enriched in the C1 sub-group; a C1 tumour is 4.9× more likely to be squamous-like than columnar-like or unclassified (Fisher Exact Test, *p* = 0.0003). This, taken together with the enrichment of alpha-7 HPV types in C2 tumours suggests that although they are SCCs (confirmed by examination of 75 cases from TCGA's digital slide archive by two pathologists blinded to TCGA-assigned histology, with close agreement (Cohen's Kappa = 0.89, 95% CI = 0.77−1); Supplementary Data 12), they harbour features associated with adenocarcinoma; possibly even hinting at a different cell-of-origin for C1 versus C2 tumours. We note that evidence from mouse models

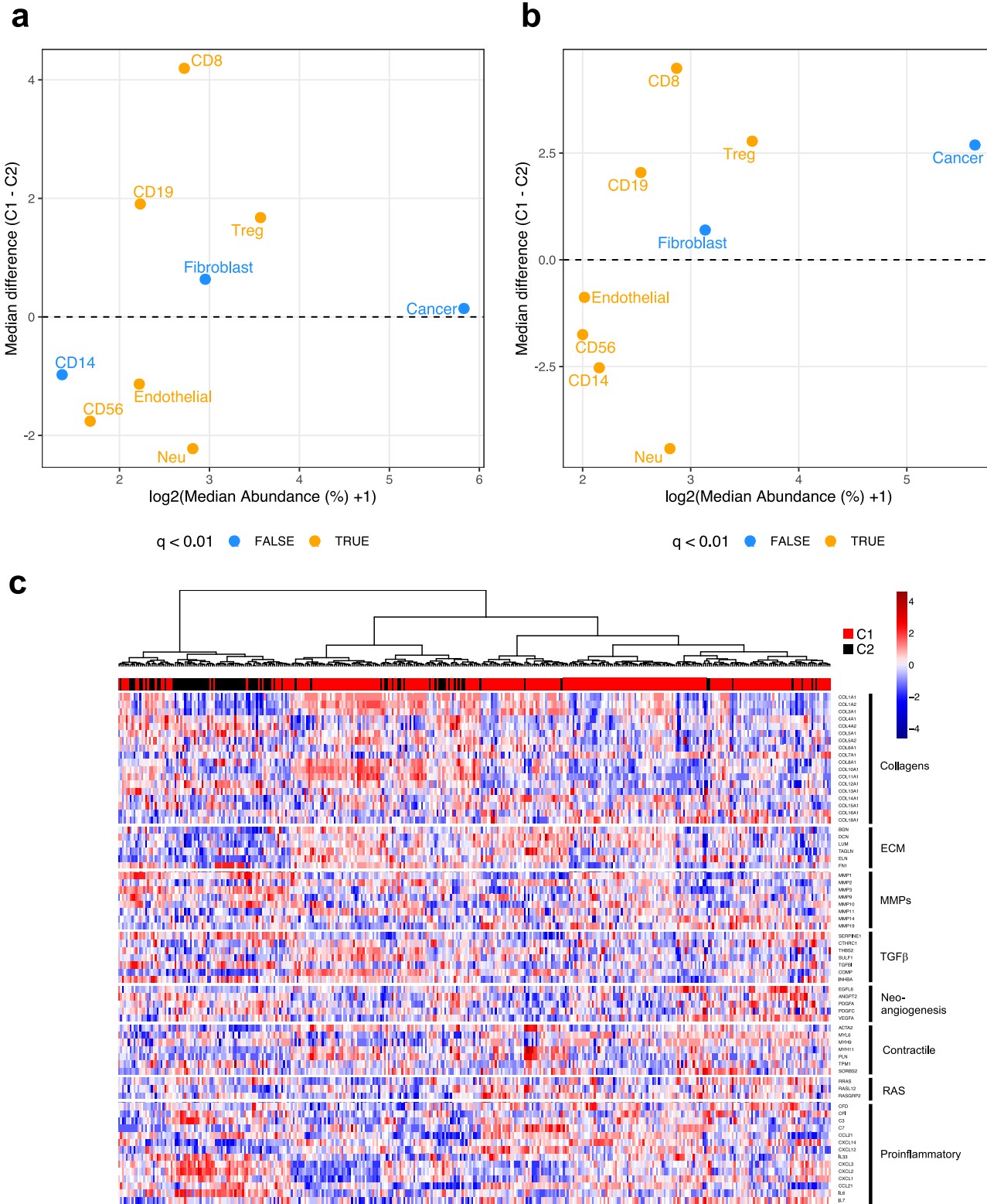

**Fig. 6 | Differences in the tumour microenvironment between cervical cancer subgroups.** Plot showing median abundances (x-axis) and median differences (%, y-axis) for different cell types estimated using MethylCIBERSORT, with significant differences (FDR < 0.01, Wilcoxon Rank Sum Test) in orange, for **a** TCGA cohort and **b** combined validation cohorts. **c** 68 genes in genesets associated with different CAF phenotypes were used to cluster patients for which RNA-seq data was available (TCGA, Bergen and Uganda). C2 tumours cluster together with high expression in proinflammatory genes, indicative of the iCAF phenotype. Source data are provided as a Source Data file.

has implicated *Lkb1* (*STK11*) loss as a key event in enabling transition from lung adenocarcinoma to lung SCC; a transition that has been linked to drug resistance in lung cancer patients[76,77]. The frequent mutation and/or loss of *STK11* in C2 tumours observed in our study suggests a similar adeno-SCC transition may be similarly linked to poor prognosis in cervical cancer. Furthermore, patients with Peutz-Jehgers Syndrome (PJS, an inherited tumour predisposition syndrome caused by germline *STK11* mutations) are at significantly increased risk of developing Minimal Deviation Adenocarcinoma, a rare and aggressive cervical malignancy that is not linked to HPV infection and sporadic cases of which frequently harbour somatic *STK11* mutations[78–80].

Our analysis suggests differences in the tumour immune microenvironment between C1 and C2 CSCCs, that are highly reproducible across cohorts from the USA, Europe and Uganda and that might explain the differential prognosis associated with these clusters. In addition to the high neutrophil:lymphocyte ratio, the increased expression of cytokines including IL-6, TGF-β and G-CSF and of the chemokines CXCL1-3 in C2 tumours suggests pro-tumourigenic (N2) polarisation of these neutrophils[81–86], which is typical of tumours with a high NLR[87]. IL-6 has also been implicated in the polarisation of macrophages towards a tolerogenic M2 state in cervical cancer[88]. The observation that CSCCs occurring in HIV+ patients from the Ugandan/ CGCI cohort are much more likely to be of the C2 subtype than those in HIV− patients hints at a possible relationship between the immune competence of the patient and the likelihood of developing a C2 tumour. This requires further investigation but is consistent with greater evidence of existing anti-tumour immune responses in C1 tumours.

Finally, we hope the identification of 21 SMGs in CSCC will stimulate functional studies on these genes and their role in cervical cancer pathogenesis, potentially enabling identification of new therapeutic targets. The identification of C2-specific alterations to YAP1 and upstream Hippo signalling pathway components is of particular interest, given recent studies that highlight the importance of this pathway in cervical carcinogenesis[89–92]. We note that three targetable immune checkpoint proteins (B7-H3, NT5E and PD-L2) are expressed at higher levels in C2 tumours. In addition to its immune suppressive effects, B7-H3 has been linked to key processes that are upregulated in these tumours including EMT and angiogenesis, through the activation of NF-κB signalling and the downregulation of E-cadherin expression[93,94]. Interestingly, the expression of B7-H3 and NT5E on CAFs has been linked to poor prognosis in gastric and colorectal cancer, respectively[48,95]. Also of relevance given our observation of differing CAF phenotype between clusters is the report that a CAF subtype (CAF-S1) identified in breast cancer that displays high levels of B7-H3 and NT5E expression is seen in tumours with low levels of CTL infiltration[96]. PD1/PD-L1 immune checkpoint blockade (pembrolizumab) was recently FDA-approved for first-line treatment of metastatic cervical cancer in combination with chemotherapy in patients whose tumours express PD-L1[97,98], while CTLA4 blockade (ipilimumab) has also shown promising activity, both as a single agent[99,100] and in combination with PD1 blockade (nivolumab)[101]. Efficacy of PD1 blockade in cervical cancer has been linked to the presence of a CD8 + FoxP3 + CD25 + T-cell subset[102] and an important limitation of our study is the inability to differentiate between CD8 + T-cell phenotypes. Nonetheless, identification of alternative, targetable immune checkpoint molecules in C2 tumours provides a potential therapeutic strategy for a subset of cervical cancers that respond poorly to chemoradiotherapy and that, given their low overall levels of T-cell infiltrates, are maybe less likely to respond to PD1 blockade than C1 tumours. It may therefore be informative to explore use of the C1 / C2 classification as a predictor of response to pembrolizumab, and within the context of ongoing clinical trials with other agents.

In conclusion, we show that CSCCs can be categorised into two subtypes, C1 and C2, among which C1 tumours have a more favourable outcome. Although HPV16 is more likely to cause C1 tumours and HPV18 C2 tumours, HPV type is not an independent predictor of prognosis, suggesting it is the tumour type rather than the causative HPV type that is critical for the disease outcome. While we acknowledge there is an urgent need for more molecular data from parts of Africa, Asia and South America, where the majority of the disease burden now lies, it is notable that the key molecular and cellular characteristics of C1 and C2 tumours are consistent among cohorts included in this study from the USA, Europe, and Sub-Saharan Africa. This suggests that the findings and underlying principle: that CSCC can develop along two trajectories associated with differing clinical behaviour that can be identified using defined gene expression or DNA methylation signatures, are of broad relevance and that they may guide improved clinical management of cervical cancer patients.

## Methods
### Patient samples
All patients gave written, informed consent before inclusion. Samples from Bergen were collected in a population-based setting from patients treated at the Department of Obstetrics and Gynaecology, Haukeland University Hospital, Bergen, Norway, from May 2001 to May 2011. The study has been approved by the Regional Committee for Medical Research Ethics in Western Norway (REK 2009/2315, 2014/ 1907 and 2018/591). For more details on sample collection see[13,103]. Samples from Innsbruck were collected and processed at the Department of Obstetrics and Gynaecology of the Medical University of Innsbruck. The study was reviewed and approved by the Ethics committee of the Medical University of Innsbruck (reference number: AN2016-0051 360/4.3; 374/5.4: 'Biobank study: Validation of a DNA-methylation based signature in cervical cancer') and conducted in accordance with the Declaration of Helsinki. Samples from Oslo (*n* = 268) were collected from patients participating in a previously published prospective clinical study[104] approved by the Regional Committee for Medical Research Ethics in Southern Norway (REK no. S-01129). Limited quantities of patient tumour samples and extracted DNA may remain and the distribution of these materials is subject to ethical approval at the institutions from which they were collected. Note that the cases in the Oslo cohort were not treated with surgery. The samples used for molecular analysis were diagnostic biopsies from the primary tumour. In all other cases, specimens were from resections of the primary tumour. Those interested in working with these samples should contact the authors to discuss their requirements. Tumours in the Oslo and Bergen cohorts were staged according to the 2009 FIGO staging system for cervical cancer and tumours in the Innsbruck cohort were staged according to the FIGO staging system valid at the time of diagnosis (1989–2010).

### Patient follow-up
Follow up of Oslo patients consisted of clinical examination every third month for the first two years, twice a year for the next three years, and once a year thereafter. When symptoms of relapse were detected, MRI of pelvis and retroperitoneum, and X-ray of thorax were performed. For the Innsbruck cohort, a total of 29 patients with invasive cervical squamous cell cancer (age 22–91 years; median 50 years), all treated between 1989 and 2010 at the Department of Obstetrics and Gynecology, Medical University of Innsbruck, were included in this study. The median observation time of all patients was 3.65 years (1331 days) (range = 0.06–21.54 years; 21–7683 days) with respect to relapse-free survival. All patients were monitored in the outpatient follow-up program of the Department of Obstetrics and Gynecology, Medical University of Innsbruck. For the Bergen cohort, that were part of a prospective study, clinical and follow-up data for the Bergen cohort, which included patient age and diagnosis, clinical tumour size, FIGO stage and disease-specific survival were collected by review of patient records and from correspondence with responsible gynaecologists if

follow-up data were continued outside of the hospital. Further information on this cohort can be found in the original study[103]. Of the 313 total European cohort patients, nine were lost to follow up with a median of 13.7 years (5012 days; range = 0.06–23.16 years; 21–8455 days) to date last seen.

## Processing of RNAseq data

Where necessary, SRA files and were converted to fastq files using SRA-dump from the SRA Toolkit (http://ncbi.github.io/sra-tools/). Kallisto[105] v0.46.1 was then used to quantify expression of GENCODE GrCh37 transcripts, repbase repeats and transcripts from 20 different high-risk HPV types with bias correction. Transcripts were combined for gene level expression quantification using R package tximport v1.3.9.

## Generation of 450k methylation profiles

100 ng DNA was bisulphite converted using the EZ DNA Methylation kit (Zymo Research) as per manufacturer's instructions. Bisulphite converted DNA was hybridised to the Infinium 450 K Human Methylation array at UCL Genomics and processed in accordance with the manufacturer's recommendations.

## Processing of DNA methylation data

IDAT files from Illumina Infinium 450k or EPIC arrays were parsed using minfi[106] v1.34 and were subjected to Functional Normalisation[107], followed by BMIQ-correction[108] for probe type distribution (which was performed for all methylation data).

## HPV typing

Assignment of HPV type to TCGA and Bergen samples using RNA-seq data was performed using VirusSeq[109]. HPV16 or 18 was detected in 208 samples from the Oslo cohort by PCR, using the primers listed in[110]. The PCR products were detected by polyacrylamide gene electrophoresis or the Agilent DNA 1000 kit (Agilent Technologies Inc, Germany). Samples from the Innsbruck cohort and the remaining non-HPV16/18 samples from the Oslo cohort (n = 40) were HPV-typed by DDL Diagnostic Laboratory (Netherlands) using the SPF10 assay, in which a PCR-based detection of over 50 HPV types is followed by a genotyping assay (LIPA$_{25}$) that identifies 25 HPV types (HPV 6, 11, 16, 18, 31, 33, 34, 35, 39, 40, 42, 43, 44, 45, 51, 52, 53, 54, 56, 58, 59, 66, 68/73, 70 and 74). If more than one HPV type was identified in a sample (e.g., HPV16 and HPV18), that sample was designated "Other" as HPV type in the study. HPV type data for the remaining samples were published previously[10,11,13].

## Prognostic analyses and tumour clustering

Only squamous cell carcinomas were considered in this study to avoid confounding from histology. Multidimensional visualisation of the molecular differences in histology was performed using Rtsne v0.16 R package with parameters available in Supplementary Data 13, and the top 10% most variable genes using mean absolute deviation after pre filtering of low count genes (n = 1385). Final cohort numbers and summaries are shown in Table 1. Unsupervised consensus clustering was performed on TCGA SCC samples using r package ConsensusClusterPlus v1.54.0. After prefiltering of genes to remove those with low read counts (75% samples read count < 1), only the top 10% most variable genes using mean absolute deviation were considered for clustering (n = 1385). 80% of tumours were sampled over 1000 iterations using all genes. PAM clustering algorithm was used and clustering distance was measured using Pearson's correlation. An optimum number of clusters (K) of 2 was obtained by using the proportion of ambiguously clustered pairs (PAC) using thresholds of 0.1 and 0.9 to define the intermediate sub-interval. PAC was used as it accurately infers K[111]. Limma-voom on RNAseq data and limma on BMIQ and Functionally-normalised 450k and EPIC data were used to identify differentially expressed genes (DEGs, FDR = 0.01, FC > 2) and methylation variable positions (MVPs, FDR = 0.01, mean delta-Beta > 0.25) between the 2 clusters, C1 and C2. The 116 MVPs (Supplementary Data 14) common to the 450k and EPIC arrays were used to allocate clusters for the Ugandan cohort. The mean delta-Beta threshold for MVPs was determined as it delivered the highest concordance between DEG and MVP signature cluster allocation in the Bergen cohort (89.5%) and high concordance in the Ugandan cohort (91.5%). The caret v6.0-93 and limma v3.52.2R packages were used to develop an SVM using 5 iterations of 5-fold Cross-Validation using DEGs and MVPs to allocate RNAseq samples in Ugandan and Bergen cohorts, 450k samples in Bergen, Innsbruck and Oslo cohorts and EPIC samples in Ugandan cohort to these subgroups. Multidimensional visualisation using R package Rtsne was performed on the TCGA and European cohorts with available DNA methylation data combined using the 129 MVPs and parameters as shown in Supplementary Data 12.

Samples from our validation cohort, comprise of cases from three European centres (Bergen and Oslo in Norway and Innsbruck, Austria) and one African centre (Uganda) were binned into these categories, and were used for subsequent statistical analyses to identify genomic and microenvironmental correlates. Survival analyses of epigenetic allocations were carried out using Cox Proportional Hazards regression with age, tumour stage, HPV type, and with surgery, radiotherapy and chemotherapy (given/not given) as covariates. R packages used were survival v3.4-0 and survminer v0.4.9. For all clinical analyses, stages were collapsed into Stages I, II, III and IV.

RNAseq data for Bergen and Ugandan samples, Illumina HumanWG-6 v3 microarray data for 137 of the Oslo samples and Illumina HumanHT-12 v4 microarray data for 109 of the Oslo samples were used to explore cluster allocation concordance accuracy between DEG and MVP signature cluster allocation. ROC curve and ssGSEA analysis were performed using R.

## Previous study comparison

140 TCGA samples from the core set analysis (TCGA, 2017) were present in our TCGA SCC cohort. Previous cluster analysis by TCGA (2017) and Chumduri et al. (2021) was compared with our C1 and C2 cluster allocation. Review of TCGA-assigned histology for 75 evaluable cases from the digital slide archive (https://cancer.digitalslidearchive.org) was performed independently by two pathologists (J.M. and G.J.T), blinded to TCGA assignments.

## Pathway analyses

Pathway and gene sets were analysed with Metascape[51]. Settings used were minimum gene set overlap of 10, p value cutoff of 0.01 and minimum enrichment of 1.5. All functional set, pathway, structural complex and miscellaneous gene sets were included in the analysis. Only hits with an FDR of less than 0.05 were included in final results.

## Mutational analyses

For TCGA data, mutation calls were obtained from SAGE synapse as called by the MC3 project. Mutations for the Bergen cohort were obtained from[13]. Ugandan mutation calls were obtained from National Cancer Institute's Genome Data Commons Publication Page at https://gdc.cancer.gov/about-data/publications/CGCI-HTMCP-CC-2020. VCFs obtained for the Ugandan cohort samples were converted to maf files using vcf2maf v1.6.21, filtered for whole-exome mutations only, and combined. Significantly mutated genes (SMGs) were identified using dNdScv[18] individually for the three cohorts. Hypermutated samples (>600 mutations[10]) were excluded from this analysis. A weighted approach was used to combine p values for each gene for the three cohorts. R package metapro[19] v1.5.8 function wFisher was used to perform this task. Genes were considered SMGs if after FDR correction of combined p values, q < 0.1. Analysis was repeated for only C1 and C2 samples individually. Two genes were removed from our list. *MUC4*

was removed due to the large size of the gene and *GOLGA6L18* was removed as this gene and its aliases were not recognised by R package maftools[112] v2.6.05.

R package maftools[112] was used to produce an oncoplot, which shows mutation rates and significance level for each SMG, calculating tumour mutational burden for individual samples, SMG mutation frequency and mutational signatures for the combined cohorts. Binomial GLMs were used to estimate associations between C1 and C2 clusters and SMG mutation frequencies.

The estimated exposures of each sample to the identified mutational signatures were calculated using R package mutsignatures[113] v2.1.1 and converted to proportion of signature exposure per sample.

### Copy number analysis

450k total intensities (Methylated and Unmethylated values) were used to generate copy number profiles with normal blood samples from Renius et al.[114] as the germline reference. Functional normalisation[107] was used to regress out technical variation across the reference and tumour datasets before merging and quantile normalisation was used to normalise combined intensities followed by Circular Binary Segmentation as previously described[115]. Median density peak correction was performed to ensure centring before further analysis. GISTIC2.0[116] was then used to identify regions of significant copy number change at both arm and gene levels. Candidate copy number changes were evaluated for association with cluster using binomial GLMs. The parameters chosen were a noise threshold of 0.1 with arm-level peel off and a confidence level of 0.95 was used to nominate genes targeted by copy number changes. Binomial regression was finally used to estimate rates of differential alteration.

### Reverse Phase Protein Assay analysis

Reverse Phase Protein Assay (RPPA) data for the core TCGA CESC samples were obtained from the NCI GDC Legacy Archive. Differentially expressed proteins between C1 and C2 clusters were determined using R package limma v3.52.2 (FDR = 0.05, FC > 1.3).

### Tumour microenvironment analyses

MethylCIBERSORT[42] was used to estimate tumour purity and abundances of nine other microenvironmental cellular fractions using TCGA and validation cohort methylation beta values. Fraction numbers were then normalised by cellular abundance and differences between clusters C1 and C2 were estimated using Wilcoxon's rank sum test with Benjamini–Hochberg correction for multiple testing. This analysis was performed separately on TCGA cohort and combined validation cohort, as well as on each individual cohort.

Cancer-associated fibroblast associated gene set lists were obtained from Qian et al.[64]. TCGA, Bergen and Ugandan cohort sample RNAseq data was combined and visualised for these gene set genes using R package NMF[117] v0.24.0.

### CAF index calculation

For cohorts for which RNAseq data were available (TCGA, Bergen and Uganda), a CAF index was calculated as described in Ko et al.[56]. The median CAF index value was used as a threshold to allocate high or low CAF in tumour samples.

### Immunohistochemistry

Immunohistochemical staining of samples from the Innsbruck cohort was conducted by HSL-Advanced Diagnostics (London, UK) using the Leica Bond III platform with Leica Bond Polymer Refine detection as per manufacturer's recommendations. Sections from a series of 17 tumour samples from the validation cohort were stained for CD8 (mouse monoclonal 4B11, Leica Biosystems PA0183, used as supplied for 15 min at room temperature. HIER was performed on-board using Leica ER2 solution (high pH) for 20 min), or myeloperoxidase (MPO,

rabbit polyclonal, Agilent A039829-2, used at a dilution of 1/4000 for 15 min at room temperature without epitope retrieval. Scoring was performed blinded to cluster membership by a histopathologist (J.M.) as follows: 0 = no positive cells/field (200× magnification); 1 = 1–10 positive cells; 2 = 11–100 positive cells; 3 = 101–200 positive cells; 4 = 201–300 positive cells; 5 = over 300 positive cells.

For the Oslo cohort, manual CD8 staining was conducted using the Dako EnVision™ Flex+ System (K8012, Dako). Deparaffinization and unmasking of epitopes were performed using PT-Link (Dako) and EnVision™ Flex target retrieval solution at a high pH. The sections were incubated with CD8 mouse monoclonal antibody (clone 4B11, 1:150, 0.2 µg IgG$_{2b}$/ml) from Novocastra (Leica Microsystems, Newcastle Upon Tyne, UK) for 45 min. All CD8 series included positive controls. Negative controls included substitution of the monoclonal antibody with mouse myeloma protein of the same subclass and concentration as the monoclonal antibody. All controls gave satisfactory results. CD8 pathology scores were given to each sample (blinded to cluster membership) for connective tissue only, tumour only and both as follows: 0 = no positive: 1 = <10% CD8 positive cells; 2 = 10–25% CD8 positive cells; 3 = 25–50% CD8 positive cells; 4 = >50% CD8 positive cells. For digital quantification scanned images of all sections at a high resolution of 0.46 um/pixel (20×), which was reduced to 0.92 um/pixel for analysis, were used. Digital score was calculated by quantifying the area fraction of stained CD8 cells in relation to the entire section in the digital assessment, as described previously for pimonidazole staining[118]. For a detailed description of this method together with representative staining images, please see Supplementary Methods.

### Statistics and reproducibility

Statistical tests were performed in R and in all cases were two-sided. Multiple test correction was performed using the p.adjust function in R base package and the Benjamini–Hochberg[119] method where appropriate. Individual statistical tests are stated in methods and figures.

Supporting R packages used included dplyr v1.0.9, matrixStats v0.52.2, NMF v0.24.0, DescTools v0.99.45, ggplot2 v3.3.6, reshape v0.8.9, GGally v2.1.2, Coin v1.4-2, ccaPP v0.3.3, magrittr v2.0.3, DESeq2, reshape2 v1.4.4, multtest v2.24.0, amap v0.8-18, dynpred v0.1.2, gridExtra v2.3, stringr v1.4.1, tidyr v1.2.0, broom v1.0.0, ggrepel v0.9.1, gtsummary v1.6.1, TxDb.Hsapiens.UCSC.hg19.knownGene v3.1.2, tximport v1.3.9, EnsDb.Hsapiens.v75 v2.99.0, mutsignatures v2.1.1, BSgenome.Hsapiens.UCSC.hg19 v1.3.1000, and dependencies.

### Reporting summary

Further information on research design is available in the Nature Research Reporting Summary linked to this article.

### Data availability

All data generated for this study are publicly available. The DNA methylation (Illumina Infinium 450k array) data generated in this study have been deposited in the Gene Expression Omnibus under accession code GSE211668. TCGA CESC DNA methylation (Illumina Infinium 450k array) and RNAseq data are available from the TCGA data portal [https://www.cancer.gov/about-nci/organization/ccg/research/structural-genomics/tcga]. TCGA mutation data are available from the MC3 project on SAGE Synapse (syn7214402), RNAseq data for the Uganda cohort are available from the TCGA data portal and DNA methylation (Illumina Infinium EPIC array) and mutation data from National Cancer Institute's Genome Data Commons Publication Page [https://gdc.cancer.gov/about-data/publications/CGCI-HTMCP-CC-2020]. DNA methylation (Illumina Infinium 450k array) and gene expression (Illumina HumanHT-12 V4.0 expression beadchip) data from the Oslo cohort are available from the Gene Expression Omnibus under accession code GSE68339. RNAseq data were obtained for the Bergen cohort from dbGaP (phs000600/DS-CA-MDS 'Genomic Sequencing of Cervical Cancers' [https://www.ncbi.nlm.nih.gov/gap/]

under the authorisation of project #14589 "Investigating the mechanisms by which viruses and carcinogens contribute to cancer development". Source data are provided with this paper.

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

## Acknowledgements

This work was supported by the following research grants and awards: Debbie Fund (UCLH Charity fund 1427, TRF and KC); a UCL postgraduate research scholarship (AC); Rosetrees Trust (M229-CD1), Cancer Research UK (A25825), the Biotechnology and Biosciences Research Council (Grant Ref: BB/V010271/1), the Royal Society (IEC\R2\202256) and the Global Challenges Doctoral Centre at the University of Kent (TRF); MRC (MR/M025411/1), PCUK(MA-TR15-009), BBSRC (BB/R009295/1), TUF, Orchid and the UCLH BRC (AF). Results shown here are

in part based upon data generated by the TCGA Research Network: https://www.cancer.gov/tcga and the Cancer Genome Characterization Initiative: https://ocg.cancer.gov/programs/cgci. Special thanks to the late Dr Helga Salvesen, a wonderful collaborator and colleague who played a key role in the project.

## Author contributions

A.C., I.R., H.S., K.C., A.F., T.R.F. designed the study; A.C., I.R., S.H., C.D., N.K., M.J., D.R.R., N.G.M., J.M., X.S., N.E., C.S.F., V.E.S., M.K.H., C.K., A.S., S.S., G.J.T., A.F., T.R.F. conducted experiments; C.K., H.L., H.F., M.L., J.D. provided unpublished data and/or reagents; P.J.I.E, M.W., M.M., J.D., H.F., H.L., K.C., T.R.F. supervised work; A.C., I.R., G.J.T., C.K., H.L., H.F., K.C., A.F., T.R.F wrote the paper.

## Competing interests

The authors declare no competing interests.

## Additional information

Ankur Chakravarthy[1,17], Ian Reddin [2,17], Stephen Henderson [3], Cindy Dong[4], Nerissa Kirkwood[4], Maxmilan Jeyakumar[4], Daniela Rothschild Rodriguez [4], Natalia Gonzalez Martinez [4], Jacqueline McDermott[5], Xiaoping Su[6], Nagayasau Egawa[7], Christina S. Fjeldbo[8], Vilde Eide Skingen [8], Heidi Lyng [8,9], Mari Kylleso Halle[10], Camilla Krakstad [10], Afschin Soleiman[11], Susanne Sprung[12], Matt Lechner[5], Peter J. I. Ellis [4], Mark Wass [4], Martin Michaelis [4], Heidi Fiegl[13], Helga Salvesen[10], Gareth J. Thomas [2], John Doorbar[7], Kerry Chester [5] ✉, Andrew Feber [14,15] ✉ & Tim R. Fenton [2,4,16] ✉

[1]Princess Margaret Cancer Centre, University Health Network, Toronto, ON, Canada. [2]Cancer Sciences, Faculty of Medicine, University of Southampton, Southampton, UK. [3]UCL Cancer Institute, Bill Lyons Informatics Centre, University College London, London, UK. [4]School of Biosciences, Division of Natural Sciences, University of Kent, Canterbury, UK. [5]UCL Cancer Institute, University College London, London, UK. [6]MD Anderson Cancer Center, Houston, TX, USA. [7]Department of Pathology, University of Cambridge, Cambridge, UK. [8]Department of Radiation Biology, Oslo University Hospital, Oslo, Norway. [9]Department of Physics, University of Oslo, Oslo, Norway. [10]Department of Obstetrics and Gynaecology, Haukeland University Hospital, Bergen, Norway; Centre for Cancer Biomarkers, Department of Clinical Science, University of Bergen, Bergen, Norway. [11]INNPATH, Institute of Pathology, Tirol Kliniken Innsbruck, Innsbruck, Austria. [12]Institute of Pathology, Medical University of Innsbruck, Innsbruck, Austria. [13]Department of Obstetrics and Gynaecology, Medical University of Innsbruck, Innsbruck, Austria. [14]Centre for Molecular Pathology, Royal Marsden Hospital Trust, London, UK. [15]Division of Surgery and Interventional Science, University College London, London, UK. [16]Institute for Life Sciences, University of Southampton, Southampton, UK. [17]These authors contributed equally: Ankur Chakravarthy, Ian Reddin. ✉e-mail: k.chester@ucl.ac.uk; a.feber@ucl.ac.uk; t.r.fenton@soton.ac.uk

