## [Peer Review File · Nature Communications]

This manuscript has been previously reviewed at another journal that is not operating a transparent peer review scheme. This document only contains reviewer comments and rebuttal letters for versions considered at *Nature Communications*.

REVIEWER COMMENTS

Reviewer #1 (Remarks to the Author): cervical cancer and HPV immunology

NCOMMS

The authors present a multi-omic analysis of 643 squamous cervical cancers. Data from the TCGA database (n = 236) were used as a discovery set, and the remaining 407 cases were queried in the validation group. Using 5-year overall survival (OS) as an outcome endpoint, the cases were segregated into two clusters, C1 (better prognosis) and C2 (worse prognosis). A few clarifications in the main text would help in discerning the clinical significance of this proposed classification.

Line 55: while technically, the authors did evaluate a total of 643 tumors, it should specify that 236 cases from TCGA were used as a discovery group, and the remaining 313 were used to carry out validation analyses.

- Table 1: is internally inconsistent. The cases from the Oslo cohort total to 270, not 248.

- Table 1: In the Oslo cohort, 21 cases of Stage I disease are noted. However, according to the treatment table, no patients were treated with surgery. While non-surgical treatment options may be considered for Stage I cervical cancers, no Stage 1 patients underwent therapeutic excision is not consistent with the current standard-of-care.

- Similarly, in the Oslo cohort, there were a total of 247 cases of Stage 2 or greater disease. The standard of care involves a combination of platinum-based chemotherapy with radiation. Please provide the justification for treating 47 patients with radiation alone? What stage disease was treated with RT alone?

Line 224: the nomenclature is the same for both your clusters and the TCGA clusters, but in reality designate different groups. Perhaps you could use 'keratin-high' instead of TCGA C1, and 'keratin low' to designate your C2. See Line 229; which C2 tumors are you discussing?

Paragraph beginning at Line 239: I think that you have to segregate your findings between the discovery samples and the validation samples. If you report findings from the discovery group, and then include those findings in your validation set, I think that you are reporting the same data as a subset of the validation set. I recommend the involvement of a biostatistician.

Line 205: "There is no difference in the breakdown of C1 and C2 tumors by stage" This statement is backwards. What clinicians want to know is, stage for stage, does the designation of C1 vs C2 provide prognostic information, and how does it improve on what is currently used to identify risk?

C1 and C2 do not appear to be prognostic, independent of other variables. For example, in the 215 Stage II tumors included in the validation cohort the mean 5-year survival was 3.54 among C1 tumors, while the mean 5-year survival of C2 tumors was 3.16.

Methods:

Immunohistochemistry

FFPE (?) slides from two cohorts, Innsbruck and Oslo, were evaluated by IHC. The problem is that the two groups were not analyzed in the same way.

The 17 samples from the Innsbruck group were stained for CD8 and CD68. "Scoring was performed...as follows: 0 = no positive cells/200x field (how many?), 1 = 1-10 positive cells, etc." Were these counts segregated by tumor and stroma? What happened with the macrophage counts?

The Oslo group samples (n = ?) were stained for CD8, (All controls gave satisfactory results) (Photomics would be useful as a reference) CD8 pathology scores were given to each sample for the intensity of CD8 infiltrates in connective tissue only, and in tumor only.

It is really not clear how to compare these or use these two datasets, or their relationship with C1 or C2 or any other metric.

While the methods appear to be presented in Lines 333-338, no statements regarding their association with any of the analytic subgroups, let alone by C1/C2.

In the end, while CSCC can indeed develop into either treatment-response vs treatment-resistant, it is not clear from this manuscript how the distinction between C1 and C2 is predictive of clinical behavior.

Reviewer #3 (Remarks to the Author): expertise in HPV virus evolution and pathology in cancer

This manuscript presents an analysis of cervix cancer TCGA multiomic data to evaluate 2 clusters they have identified for cancer outcome and survival. Improvements in classification of cancer that could distinguish different forms that might seem similar at the clinical or histologic level, but are different at the molecular level is of interest. Nevertheless, the manuscript has a number of shortcomings and deficiencies that need to be addressed. (1) Foremost, the description of the outcome – long term follow-up of cancer cases could use a more epidemiological approach. For instance, there is no mention of loss to follow-up, how was follow-up ascertained, etc. TCGA was initially designed to describe genetic changes in cancer, thus tumors that were larger in size would be more likely to be included, as would samples that could be surgically removed. This type of information needs to be in the manuscript and the authors should assess the exact starting cohorts, etc. (2) The way the study was initially designed could lead to “immortal time bias” in the variables associated with survival. (3) If this data is relevant to all cervical cancer, why exclude adenocarcinomas? This is a critical group to keep in the analyses, in fact they might demonstrate that in fact, cluster 2 is really adenocancers at the molecular level (4) Given the HPV type differences in the cluster 1 and cluster 2 groups, a blinded analysis of the histology should be performed to determine if there was misclassification of cluster 2. As the author’s state, they seem to resemble glandular cancers and perhaps the histology was not so clear-cut.

Abstract – cervical cancer is a leading cause of cancer deaths in women.

Line 81/82- HPV16 is also more common in adenocarcinoma, however the ratio of HPV16/HPV18 is vastly different.

Table 1 needs footnotes. Percentages should be presented for stage, HPV type, HPV clade, etc.

Define TSNE.

Include statistical significance when reporting comparison of analyses, e.g., lines 180-182; 305-7, etc.

Table 2 needs more information. What are the variables being analyzed? What was controlled for, etc.?

Why were the hypermutated samples excluded? They constitute ~9% of the samples.

Figure legends need to describe all aspects of what is shown in the figures. For instance, there was no description of what rows represented in the heatmaps of figures 1-3.

Reviewer #4 (Remarks to the Author): expert in computational biology and multi-omics in cancer

Second referee’s report on “Integrated analysis of cervical squamous cell carcinoma cohorts from three continents reveals conserved subtypes of prognostic significance.”

Background: In April 2020, I was part of a group of three referees who reviewed a very different version of this manuscript entitled "Pro-metastatic gene expression, immune evasion and an altered HPV spectrum characterize an aggressive subtype of cervical cancer" for Nature Cancer. We were reviewer 2. The manuscript was rejected and I think it is fair to judge that the concerns of reviewers 1 and 3 about the April 2020 submission were overlapping with but more serious than our concerns. Although 20 months have passed, the main conclusions of the new manuscript are different, the title is completely different, the author list is different, etc., the Associate Editor, Dr. McGinnis has asked me to treat this submission as a revised manuscript and to focus this report on the answer to the question:

"Have your concerns been addressed in the revisions?"

My general answer is: In the rebuttal letter, the authors attempted to address our concerns. The manuscript is much improved. The writing is very good except for some figure legends that are too terse.

However, the authors failed to check carefully that everything they claimed to do in the rebuttal letter is actually done in the revised manuscript. I suspected this might be a problem because the rebuttal letter does not follow the widely used style that when a response/revision entails an insertion spanning one paragraph or less, then the inserted manuscript piece is quoted verbatim in the letter. Following below are two irrefutable examples from our comments and I believe there are other examples among the responses to reviewers 1 and 3:

In the rebuttal letter, the authors wrote: "We do however, acknowledge in our discussion, the need for more molecular profiling of cervical cancers from Africa, South America and Asia, where the vast majority of the cervical cancer burden falls." However, this issue is not discussed in the revised Discussion.

In our initial report, we suggested 10 references that should be cited because they provide(d) very relevant background information. In the rebuttal letter, the authors wrote gratefully that "We thank the reviewers for directing us to these key papers, several of which we have now cited in the revised submission and we apologise for the previous omissions." However, none of the suggested background papers are actually cited in the revised paper. I am not an author of any of these background papers, so my concern was and remains purely that the authors do not provide the background information as thoroughly as they could do to put their new findings in context.

Overall, otherwise, I am generally satisfied that the authors' new methods of analysis are rigorous. I am also satisfied that their new finding of two molecularly defined clusters of squamous cell cervical carcinoma with different overall survival is potentially important. The two clusters do not (yet) suggest different treatments but having information about different prognosis is still useful.

The following mostly minor items should be clarified or corrected:

1. In the legends of Figure 1b and Figure 3, it is not explained that the five rows above the main rectangle represent the cluster, the HPV type, the clade, stage, and grade. The color codes are confusing to me because, for example, my colorblind eyes perceive that the same shade of blue is being used to represent HPV16, Alpha 9 clade, Stage II, and G2 grade.

2. I do not understand the left part of Figure 4. Intuitively, how can it be that the ranking of lengths of the gray bars differs so much from the ranking by percent of tumors mutated on the right side. I understand that these two rankings will differ somewhat because the gray bars take into account the gene length and the mutation frequencies do not consider the length, but the length alone cannot explain differences of a factor of 10 in gray bar lengths. I do not understand why the possible value of a gray bar is shown as 0.27, since $10^{-0.27} = 0.53$, which is not statistically significant. It seems that the left-hand side is described at line 622 as an "oncplot" but without an explanation of how one is supposed to interpret an oncplot.

3. In Figure 5, I do not understand why the band 11q11 is shown in the two-digit representation and other bands, such as 6p22.1, are shown in the three-digit representation. It is not clear what

were the eligible CNA intervals for this analysis. When two tumors have overlapping CNAs with substantially different boundaries, it is unclear how it was decided whether these two events count as the same CNA or different CNAs. Amplifications adjacent bands 11q22.1 and 11q22.2 are depicted as distinct CNAs, but is it not the case that many tumors have a single amplification spanning both bands?

4. The Figure 6 legend fails to explain from what statistical test(s) the labels “ $q < 0.01$ ”, “false”, “true” are derived.

5. For many of the statistical tests (e.g., line 134, line 383), the authors do not indicate whether the test was one-sided or two-sided.

6. Since the authors observed that the gene STK11 (lines 279 and 286) is a frequent target of somatic (and likely heterozygous) mutations in their cohort, they should comment on the incidence of cervical among patients with Peutz-Jeghers (cancer predisposition) syndrome in which patients typically have one allele of STK11 mutated in the germline and the second allele mutated somatically in their tumors.

7. At lines 180-182, the difference in proportions between 20/25 and 39.69 is described as “interesting”, but the authors should report whether it is statistically significant by Fisher’s exact test.

8. In the references, the article titles and journal names are not formatted consistently. Some article titles have most words starting with UPPER CASE letters, while other titles have only the first word starting with an UPPER CASE letter. The same case inconsistency occurs for the journal titles that contain more than one word. Look for example at the consecutive references 97 (Genome biology) and 98 (Genome Biology). Why does reference 28 include the digital object identifier (doi)?

9. At a few places (e.g., lines 487, 518, 521, 522, 541, 561, etc.), the citations are not properly typeset as superscripts.

10. Line 667 should start “For cohorts for which RNASeq data were available...”

11. Line 685 has an equals sign that was meant to be a double dash

12. The font color changed to non-black at lines 698-701.

Reviewers' comments:

Reviewer #1 (Remarks to the Author): cervical cancer and HPV immunology

NCOMMS

The authors present a multi-omic analysis of 643 squamous cervical cancers. Data from the TCGA database (n = 236) were used as a discovery set, and the remaining 407 cases were queried in the validation group. Using 5-year overall survival (OS) as an outcome endpoint, the cases were segregated into two clusters, C1 (better prognosis) and C2 (worse prognosis). A few clarifications in the main text would help in discerning the clinical significance of this proposed classification.

We thank the reviewer for their thorough and insightful review and hope that in addressing their comments we have managed to clarify the clinical significance of our findings.

Line 55: while technically, the authors did evaluate a total of 643 tumors, it should specify that 236 cases from TCGA were used as a discovery group, and the remaining 313 were used to carry out validation analyses.

Due to space constraints in the abstract and the fact that while TCGA served as a discovery group for class discovery and epigenetic biomarker development, the analyses of genomic correlates are pan-dataset, we propose simply removing "..., totalling 643 tumours..." from the abstract. We explain in detail (Table 1 and the first results section) how the overall cohort was broken down for the different analyses.

- Table 1: is internally inconsistent. The cases from the Oslo cohort total to 270, not 248.

Thank you for pointing out this oversight. In our analysis we included only the HPV-containing tumours, reasoning that the distinct etiology of HPV-negative CESC may have confounded the derivation of a molecular classifier. Table 1 mistakenly included 22 HPV-negative tumours in the section on staging, hence the 270 total in that section. These have now been removed.

- Table 1: In the Oslo cohort, 21 cases of Stage I disease are noted. However, according to the treatment table, no patients were treated with surgery. While non-surgical treatment options may be considered for Stage I cervical cancers, no Stage 1 patients underwent therapeutic excision is not consistent with the current standard-of-care. Similarly, in the Oslo cohort, there were a total of 247 cases of Stage 2 or greater disease. The standard of care involves a combination of platinum-based chemotherapy with radiation. Please provide the justification for treating 47 patients with radiation alone? What stage disease was treated with RT alone?

In the Oslo cohort, all patients were planned for standard chemoradiotherapy with curative intent. This treatment is consistent with standard-of-care for stage IB (bulky)-IVA disease at many institutions in Western Europe. The radiotherapy consists of external radiation, followed by brachytherapy. Concomitant cisplatin is given according to tolerance. Stage IA patients are treated with surgery but the 21 cases with stage I disease in this cohort all had stage IB disease and were therefore treated with chemoradiotherapy and not surgery. If tolerated, patients receive weekly cisplatin (40mg per m²), maximum 6 courses, during the period of external radiotherapy. Some patients have to interrupt the cisplatin regime because of poor tolerance (nausea etc). These patients have fewer or no courses with cisplatin (only radiotherapy). This is standard-of-care, and the study cannot be considered as a clinical trial. There are no specific stages that are treated with RT alone, it depends on the performance of the individual patients. The patient-by-patient information regarding who received standard chemoradiotherapy and who received radiotherapy without concomitant chemotherapy is included in **Supplementary Table 6**.

Line 224: the nomenclature is the same for both your clusters and the TCGA clusters, but in reality designate different groups. Perhaps you could use 'keratin-high' instead of TCGA C1, and 'keratin low' to designate your C2. See Line 229; which C2 tumors are you discussing?

We agree that the wording in this section (in which we are discussing the similarities and differences between our clusters and the clusters identified by TCGA) was confusing and we have attempted to clarify by making the modifications highlighted in red in the text (lines 243-260). We have explained why we think it is inappropriate to use the terms 'keratin-low' and 'keratin-high' for our C1 and C2 clusters, since the differences in keratin expression profiles between C1 and C2 tumours are more nuanced than 'high' versus 'low'.

Paragraph beginning at Line 239: I think that you have to segregate your findings between the discovery samples and the validation samples. If you report findings from the discovery group, and then include those findings in your validation set, I think that you are reporting the same data as a subset of the validation set. I recommend the involvement of a biostatistician.

The issue raised here is certainly an important one and we were very careful to avoid such a mistake by taking the following precautions:

- 1) Where the validation / discovery cohort distinction matters is in the discovery of the classes and in establishing this as a robust classification. Ceccarelli, Verhaak et al's identification of glioma subtypes (Ceccarelli *et al.*, 2016) set a precedent for classification in one cohort using clustering and using a single platform for assignment of a second cohort to the classification scheme. Once all samples have been assigned to classes, correlative analyses with data types not used for the original classification is appropriate.**
- 2) Where methodological differences are present (e.g. calling pipelines for genomic data, RNAseq), we have taken additional measures to account for this. Specifically, we conducted analyses on the separate cohorts, reporting each in supplementary figures. Only in those cases where it was appropriate to do so did we subsequently combine data to achieve greater statistical power, e.g. using p-value combination for dNdSCV and looking at ssGSEA scores**

within each). It was appropriate to combine the cohorts for our analysis of DNA copy number data because they were not relevant to class discovery *per se* and were derived from comparable platforms.

Line 205: “There is no difference in the breakdown of C1 and C2 tumors by stage” This statement is backwards. What clinicians want to know is, stage for stage, does the designation of C1 vs C2 provide prognostic information, and how does it improve on what is currently used to identify risk?

We fully agree that the clinical utility of a prognostic classification such as this lies in its ability to give information above and beyond that gained from staging. Our comment here was intended to highlight the fact we have not simply discovered a gene expression-based proxy for tumour stage (i.e. a classification that designated all stage 1 tumours as C1 and all stage 3 tumours as C2 would be ‘prognostic’ while not being particularly useful). While we have included the breakdown of C1/C2 by stage in Supplementary Table 7, we note that this information is also implicit in the results of our multivariate analyses, in which tumour stage was included as a covariate (see below).

C1 and C2 do not appear to be prognostic, independent of other variables. For example, in the 215 Stage II tumors included in the validation cohort the mean 5-year survival was 3.54 among C1 tumors, while the mean 5-year survival of C2 tumors was 3.16.

In Table 2 we present the multivariate analyses of the TCGA and European cohorts with age, tumour stage, HPV type and (in the case of the European cohort), treatment all included as covariates. In both cases, C1/C2 membership is an independent predictor of survival (overall survival in the case of TCGA and disease-specific survival in the case of the European validation cohorts, where much more detailed clinical follow-up data were available). We also removed the 5-year cut-off from our survival analyses in the revised manuscript, so all events are now included. Our ability to use disease-specific survival in the combined European validation cohort is a key strength of this study – and clearly separates it from studies that have focused solely on TCGA, or on other single cohorts for which disease-specific survival is not available.

In the specific example cited by the reviewer (215 Stage II tumours, 5-year survival C1 3.54 vs 3.16 for C2), this includes the Ugandan cohort, which is not suitable for survival analysis and particularly not for combining with the European cohort for the following reasons outlined in our revised manuscript. (1) Disease-specific survival data are not available for this cohort; (2) the OS of this cohort is considerably shorter than the DSS of the combined and individual European cohorts (Supplementary Figure 5d); (3) 80% of the C2 patients and 54% of the C1 patients in the Ugandan cohort are also HIV⁺. The strong relationship between HIV status and OS observed in this study (Gagliardi *et al.*, 2020) likely masks other differences (we also note that neither HPV type nor clade are prognostic within this cohort).

Methods:

Immunohistochemistry

FFPE (?) slides from two cohorts, Innsbruck and Oslo, were evaluated by IHC. The problem is that the two groups were not analyzed in the same way.

The IHC on these cohorts was performed separately, hence the differences in methodology for staining and analysis. However, the IHC was used only for validation for our DNA methylation (MethylCIBERSORT) estimates of immune cell infiltrates and comparisons were made strictly within and not between cohorts, to facilitate robust interpretation. MethylCIBERSORT on the other hand, was run using DNA methylation data generated from microarrays that are standardised and comparable. IHC estimates are not meant to facilitate between-cohort analyses precisely to account for the variability in scoring methods between the different centres that contributed to this study.

The 17 samples from the Innsbruck group were stained for CD8 and CD68. "Scoring was performed...as follows: 0 = no positive cells/200x field (how many?), 1 = 1-10 positive cells, etc." Were these counts segregated by tumor and stroma? What happened with the macrophage counts?

We apologize, the reference to anti-CD68 (macrophage marker) staining in the methods section was included in error. The Innsbruck samples were stained for CD8 and myeloperoxidase (neutrophil marker) and the Oslo samples were stained for CD8.

The Oslo group samples (n = ?) were stained for CD8, (All controls gave satisfactory results) (Photomics would be useful as a reference) CD8 pathology scores were given to each sample for the intensity of CD8 infiltrates in connective tissue only, and in tumor only.

N=229 has now been added to this section of the Methods and to the legend for Supplementary Figure 11.

"For 229 samples from the Oslo cohort, manual CD8 staining was conducted using the Dako EnVision™ Flex+ System (K8012, Dako). Deparaffinization and unmasking of epitopes were performed using PT-Link (Dako) and EnVision™ Flex target retrieval solution at a high pH. The sections were incubated with CD8 mouse monoclonal antibody (clone 4B11, 1:150, 0.2 µg IgG2b/ml) from Novocastra (Leica Microsystems, Newcastle Upon Tyne, UK) for 45 minutes. All CD8 series included positive controls. Negative controls included substitution of the monoclonal antibody with mouse myeloma protein of the same subclass and concentration as the monoclonal antibody. All controls gave satisfactory results. CD8 pathology scores were given to each sample (blinded to cluster membership) for connective tissue only, tumour only and both as follows: 0 = no positive: 1 = <10% CD8 positive cells; 2 = 10-25% CD8 positive cells; 3 = 25-50% CD8 positive cells; 4 = >50% CD8 positive cells. For digital quantification scanned images of all sections at a high resolution of 0.46 µm/pixel (20x), which was reduced to 0.92 µm/pixel for analysis, were used. Digital score was calculated by

quantifying the area fraction of stained CD8 cells in relation to the entire section in the digital assessment.”

Could the reviewer please clarify what is meant by ‘Photomics would be useful as a reference’ here please? We are happy to provide high-res images from controls / samples in the supplementary if helpful.

It is really not clear how to compare these or use these two datasets, or their relationship with C1 or C2 or any other metric.

See above – our only intention with the IHC analysis was to examine the concordance with the DNA methylation-based estimates of immune cell infiltration within and not to make comparisons between cohorts or clusters. However we can confirm that the IHC based estimates of CD8+ T-cell infiltration are higher in C1 versus C2 tumours ($p = 0.024$, Wilcoxon Rank Sum test), consistent with the DNA methylation-based finding.

While the methods appear to be presented in Lines 333-338, no statements regarding their association with any of the analytic subgroups, let alone by C1/C2.

As above, the sole intention of the staining was to validate the MethylCIBERSORT-based estimates, which clearly indicate a greater CD8+ T-cell abundance in C1 tumours and greater neutrophil content in C2 tumours. This is true in each individual cohort. We were able to conduct CD8 staining (for CD8+ lymphocytes) on 229 samples from the Oslo cohort and staining for CD8 and for myeloperoxidase (neutrophil marker) in 14 samples from the Innsbruck cohort. The scoring from this staining was only used to check the correlation with methylation-based deconvolution (which has also been independently verified in multiple studies).

In the end, while CSCC can indeed develop into either treatment-response vs treatment-resistant, it is not clear from this manuscript how the distinction between C1 and C2 is predictive of clinical behavior.

We would contend that the independent utility of the C1/C2 classification as a prognostic factor is clear from the multivariate analyses presented. Importantly this is true not only for TCGA but also for the European cohort, which has excellent long-term disease-specific survival data and for which sufficient information on treatment was available to include it as a covariate in our Cox regression. The fact that we have shown C1/C2 to be an independent predictor of disease-specific survival in the validation cohort clearly suggests it to be a useful means by which to classify CSCC and gives significance to the biological differences between C1 and C2 tumours that we have identified in this study. Taken together, we believe these findings represent an important advance in our knowledge of cervical cancer.

Reviewer #3 (Remarks to the Author): expertise in HPV virus evolution and pathology in cancer

This manuscript presents an analysis of cervix cancer TCGA multiomic data to evaluate 2 clusters they have identified for cancer outcome and survival. Improvements in classification of cancer that could distinguish different forms that might seem similar at the clinical or histologic level, but are different at the molecular level is of interest. Nevertheless, the manuscript has a number of shortcomings and deficiencies that need to be addressed.

We thank the reviewer for their careful critique of the student and welcome their positive comments relating to the utility of molecular classifiers for tumours that appear similar at the clinical or histologic level.

- (1) Foremost, the description of the outcome – long term follow-up of cancer cases could use a more epidemiological approach. For instance, there is no mention of loss to follow-up, how was follow-up ascertained, etc. TCGA was initially designed to describe genetic changes in cancer, thus tumors that were larger in size would be more likely to be included, as would samples that could be surgically removed. This type of information needs to be in the manuscript and the authors should assess the exact starting cohorts, etc.

We fully agree with the reviewer on these points and apologise that this information was not presented more clearly in the previous version of the manuscript. Our primary motivation for curating a validation cohort in which to test inferences made from TCGA data was indeed the fact that for reasons including those highlighted by the reviewer, while TCGA is an invaluable resource for gaining molecular insight, making linkages to clinical outcome can be challenging. As detailed in our responses to Reviewer 1, we have now been able to present disease-specific survival for the combined European cohort, with only 9 of 313 patients lost to follow-up, which we feel strengthens the study considerably. We have also added detailed information on follow-up in a new Methods section (lines 548-565, copied below for reference). Table 1 contains an overview of the cohorts and Supplementary Table 6 contains detailed clinical and pathological data on a patient-by-patient basis, including detailed staging.

“Follow up of Oslo patients consisted of clinical examination every third month for the first two years, twice a year for the next three years, and once a year thereafter. When symptoms of relapse were detected, MRI of pelvis and retroperitoneum, and X-ray of thorax were performed. For the Innsbruck cohort, a total of 29 patients with invasive cervical squamous cell cancer (age 22-91 years; median 50 years), all treated between 1989 and 2010 at the Department of Obstetrics and Gynecology, Medical University of Innsbruck, were included in this study. The median observation time of all patients was 3.65 years (1331 days) (range = 0.06 – 21.54 years; 21 – 7,683 days) with respect to relapse-free survival. All patients were monitored in the outpatient follow-up program of the Department of Obstetrics and Gynecology, Medical University of Innsbruck. For the Bergen cohort, that were part of a prospective study, clinical and follow-up data for the Bergen cohort, which included patient age and diagnosis, clinical tumour size, FIGO stage and disease specific survival were collected by review of patient records and from correspondence with responsible gynaecologists if follow-up data were continued outside of the hospital. Further information on this cohort can be found in the original study. Of the 313 total European cohort patients, nine were lost to follow up with a median of 13.7 years (5,012 days; range = 0.06 – 23.16 years; 21 – 8,455 days) to date last seen.”

- (2) The way the study was initially designed could lead to “immortal time bias” in the variables associated with survival.

According to Suissa (2008) “Immortal time refers to a span of time in the observation or follow-up period of a cohort during which the outcome under study could not have occurred. It usually occurs with the passing of time before a subject initiates a given exposure.”

Our study follows cohorts of patients who were only included if they underwent treatment for their cervical cancer (and thus there was a sample of their tumour available for molecular analysis. This occurred very rapidly following the date of diagnosis (used as T_0 in all our survival analyses)). After this point, there is no time period during which the outcome under study (death due to any cause for TCGA or death due to cervical cancer for our validation cohort) could not have occurred and been recorded.

Chemo/radiotherapy is necessarily administered over a period following diagnosis (and surgery in those patients who received it). Any deaths that occurred during this time have been recorded however, therefore while we agree that a patient who died before completing a course of chemo/radiotherapy would not have received the entire treatment, this is an issue inherent in any such analysis of survival among cancer patients. We don't see how our study design could lead to immortal time bias but would be happy to explore this point further with the reviewer.

(3) If this data is relevant to all cervical cancer, why exclude adenocarcinomas? This is a critical group to keep in the analyses, in fact they might demonstrate that in fact, cluster 2 is really adenocancers at the molecular level

We are not asserting that these data are relevant to all cervical cancer; rather we have presented a focused analysis of cervical squamous cell carcinoma, the most common histological subtype of cervical cancer. Numerous studies have shown clear differences in factors including prognosis, age at diagnosis, HPV type association and genomics between cervical adenocarcinoma and cervical SCC (e.g. Li *et al.*, 2011; Jung *et al.*, 2017; Galic *et al.*, 2012; Burk *et al.*, 2017). By restricting our analysis to SCC, we have avoided the risk of ‘identifying’ molecular associations that are simply a reflection of these well-documented differences in the biology and clinical behaviour of cervical adenocarcinoma and SCC. Many studies in cervical cancer and in other cancer types for which adenocarcinoma and SCC arise in the same tissue have similarly focused on one histologic subtype, notably in lung and oesophagus. Indeed TCGA conducted separate projects on lung adenocarcinoma (LUAD) and SCC (LUSC).

Although the SCCs in these cohorts are clearly distinct from adenocarcinomas at the whole transcriptome level (Supplementary Figure 1a-c), we do agree that cluster 2 shares certain molecular features with adenocarcinoma, as evidenced by enrichment of ‘columnar-like’ (Chumduri *et al.*, 2021) tumours among C2 (Figure 3 and Discussion, lines 444-461). In our revised manuscript discussion (lines 461-466), “We note that evidence from mouse models has implicated *Lkb1* (STK11) loss as a key event in enabling transition from lung adenocarcinoma to lung SCC; a transition that has been linked to drug resistance in lung cancer patients (Han *et al.*, 2014; Hou *et al.*, 2016). The frequent mutation and/or loss of STK11 in C2 tumours observed in our study suggests a similar adeno-SCC transition may be similarly linked to poor prognosis in cervical cancer.” Proving this speculation would require a complex set of lineage tracing experiments that is beyond the scope of the current study but we hope our observation will stimulate further research on this important topic. We verified that TCGA C2 tumours are indeed SCCs using the digital image archive (now added as Supplementary Table 12) and response to point (4) below).

(4) Given the HPV type differences in the cluster 1 and cluster 2 groups, a blinded analysis of the histology should be performed to determine if there was misclassification of cluster 2. As the author's state, they seem to resemble glandular cancers and perhaps the histology was not so clear-cut.

Thank you for this suggestion – we had the same thought upon initial analysis of the molecular data. To address this, H&E-stained sections from TCGA's digital image archive were reviewed by a histopathologist (Dr. J McDermott) who was blinded to TCGA's histological designation (Supplementary Table 12). In almost all cases, our pathologist was in agreement with TCGA's designation of these as CSCCs.

"This, taken together with the enrichment of alpha-7 HPV types in C2 tumours suggests that although they are SCCs (confirmed by examination of selected images from TCGA's digital image archive by a pathologist blinded to TCGA-assigned histology, Supplementary Table S12), they harbour features associated with adenocarcinoma; possibly even hinting at a different cell-of-origin for C1 versus C2 tumours". Discussion, lines 456-461.

Abstract – cervical cancer is a leading cause of cancer deaths in women.

We've updated the abstract accordingly (replaced "... represents one of the leading causes of cancer death worldwide" with "... is a leading cause of cancer deaths in women").

Line 81/82- HPV16 is also more common in adenocarcinoma, however the ratio of HPV16/HPV18 is vastly different.

Thank you for pointing out the ambiguity in this sentence. We've re-worded this sentence to quote the global statistics from 1990-2010 directly from the meta-analysis by Li et al (reference 2).

"HPV type is also associated with histology; HPV16 and HPV18 were reported in 59.3% and 13.2% of CSCC and in 36.3% and 36.8% of adenocarcinoma respectively worldwide, between 1990 and 2010²" Introduction, lines 81-84.

Table 1 needs footnotes. Percentages should be presented for stage, HPV type, HPV clade, etc.

We've included percentages and footnotes in our revised Table 1 as suggested.

Define TSNE.

We have included this in the first paragraph of the revised results section: "...were apparent in multi-dimensional t-distributed stochastic neighbour embedding (TSNE) analysis based on the top 10% most variable genes..." Results, lines 120-122.

Include statistical significance when reporting comparison of analyses, e.g., lines 180-182; 305-7, etc.

We have now added p-values and the statistical test used to derive them in all such comparisons, see examples below:

“Interestingly 80% (20/25) of Ugandan C2 patients were human immunodeficiency virus (HIV) positive, while only 56% (39/69) of C1 patients were HIV positive (OR = 0.33; $p = 0.05$, Fisher’s Exact Test, Supplementary Fig. S5a).” Results, lines 191-193.

“There is considerable overlap between our C1 cluster and TCGA’s ‘mRNA-C2’ cluster (84/106, odds ratio (OR) = 9, $p = 3.1 \times 10^{-7}$, Fisher’s Exact Test) and keratin-high iCluster (80/106, OR = 44.7, $p = 9.8 \times 10^{-13}$, Fisher’s Exact Test), and between our C2 cluster and TCGA’s ‘mRNA-C3’ cluster (19/34, OR = 8.15, $p = 1.7 \times 10^{-6}$, Fisher’s Exact Test) and keratin-low iCluster (27/34, OR = 18.3, $p = 5.02 \times 10^{-11}$, Fisher’s Exact Test).” Results, lines 242-248.

Table 2 needs more information. What are the variables being analyzed? What was controlled for, etc.?

We’ve revised Table 2 to show that the comparisons are for C2 versus C1 and have included footnotes to indicate which covariates were included in each analysis. This information is also in the methods:

“Survival analyses of epigenetic allocations were carried out using Cox Proportional Hazards regression with age, tumour stage, HPV type, and with surgery, radiotherapy and chemotherapy (given/not given) as covariates” Methods, lines 645-648.

Why were the hypermutated samples excluded? They constitute ~9% of the samples.

Models that rely on dN/dS when estimating selection rely on accurately estimating a background mutation rate that applies to the models of somatic evolution usually seen with tumours. Hypermutation imposes specific biases meaning that combining hypermutated and typical samples can produce a biased background model that is either unsuited for hypermutated tumours, or for the bulk of the tumours since the very large number of mutations from hypermutants can have an outweighing influence. This is what motivated the standard practise of separating hypermutant and non-hypermutant tumours in the original dN/dSCV paper from the Sanger. See the specific section on hypermutators in this manuscript; it would require fitting a distinct background model based on the source of the hypermutation involved. <https://www.ncbi.nlm.nih.gov/pmc/articles/PMC5720395/>.

Figure legends need to describe all aspects of what is shown in the figures. For instance, there was no description of what rows represented in the heatmaps of figures 1-3.

We have included more detail throughout the figure (and supplementary figure) legends, as highlighted in red in the following examples:

“Figure 1 Derivation of prognostic clusters in TCGA SCC cohort. a) Consensus clustering of 236 TCGA HPV+ SCC patients produced an optimum 2 clusters (C1 and C2). Dark blue on the heat map represents patients (both row and column in heatmap) that clustered together 100% of the time while white represents patients that never clustered together after 90% sampling and replacement over 1000 iterations and considering only the 10% most variable genes. b) There were 938 differentially expressed genes between the two clusters. Expression values for each of the 938 genes is represented by the rows in the heatmap (columns represent individual patients). Kaplan-Meier curve shows that overall survival was higher in C1 patients in comparison to C2 patients when considering all patients (c). C2 is an individual predictor of prognosis, as is stage 4, while HPV type is not, as illustrated in a forest plot using cox-regression multivariate analysis using cluster type, age, stage and HPV type as covariates (d). Survival analysis using a Kaplan-Meier curve shows that overall survival was higher in C1 patients when considering only HPV16+ tumour patients (e). P-values on Kaplan-Meier curves from log-rank test, p-values in forest plot from Wald test.”

“Figure 2 Cluster allocation of validation cohorts using methylation signature. A DNA methylation based signature of 129 methylation variable positions (MVPs; $\text{dB} > 0.25$, $\text{FDR} < 0.01$), represented by rows in heatmaps, separates clusters C1 and C2 patients in the TCGA cohort ($n = 236$) (a) and the validation dataset from the 3 European centres ($n = 313$) (b). c) Using DNA methylation data (BMIQ corrected beta values) for the TCGA and European cohorts, C2 tumours cluster together based on the 129 MVP signature. d) The disease specific survival (DSS) for C1 patients is higher than for C2 patients in the combined European validation cohort. (e) C2 is an individual predictor of prognosis, as is stage 3 and 4, while HPV type is not, as illustrated in a forest plot using cox-regression multivariate analysis using cluster type, age, stage, HPV type and treatment regimen (RT = radiotherapy, CRT = chemoradiotherapy) as covariates. P-values on Kaplan-Meier curves from log-rank test, p-values in forest plot from Wald test.”

“Figure 3 Comparison of SCC subgroups with previous studies. Cluster analysis had previously been performed on 140 TCGA SCC tumours in two studies – one determined clusters based on cell of origin markers (Chumduri *et al*, 2021, red), and one determined clusters based on integrated omics data (TCGA Network, 2017, orange). Patients (columns) are identified as belonging to one of the clusters identified by Chumduri *et al* by red in the appropriate row and belonging to one of the TCGA clusters by orange in the appropriate row. C2 patients were more likely to belong to the Chumduri *et al* columnar-like cluster than the squamous-like cluster (odds ratio (OR) = 9.95, $p = 0.006$, Fisher’s Exact Test) and more likely to belong to the TCGA keratin-low iCluster (OR = 18.3, $p = 5.02 \times 10^{-11}$, Fisher’s Exact Test) than other iClusters. The heatmap at the bottom of plot represents expression levels of cytokeratin genes present in our C2 gene signature. 9 of the 14 genes exhibit low expression in C2 tumours, and 5 are highly expressed and so the TCGA “keratin-low” nomenclature was not used despite the large overlap with our C2 cluster.”

“Supplementary Figure S6 Elevation of epithelial mesenchymal transition (EMT) score is evident in C2 tumours. The EMT score derived by TCGA for 140 HPV+ squamous TCGA cervical cancer tumours is

higher in the C2 compared to the C1 subgroup in our study. Each point represents a TCGA HPV+ tumour. The upper line in a box plot represents the upper quartile, the second line the median and the lowest line the lower quartile. The whisker above the box is drawn to the highest point within 1.5x the interquartile range (IQR), the whisker below the box is drawn to the lowest point within 1.5x the IQR. The P value is from Wilcoxon's rank sum test."

"Supplementary Figure S8 Increased levels of YAP in tumours with YAP1 amplification. YAP1 expression (a), and YAP protein levels (b) unphosphorylated and c) phosphorylated are higher in tumours that contain YAP1 amplifications. Blue points represent tumours without YAP1 amplification and red points represent tumours with YAP1 amplification. The upper line in a box plot represents the upper quartile, the second line the median and the lowest line the lower quartile. The whisker above the box is drawn to the highest point within 1.5x the interquartile range (IQR), the whisker below the box is drawn to the lowest point within 1.5x the IQR. P values are from Wilcoxon's rank sum test."

Reviewer #4 (Remarks to the Author): expert in computational biology and multi-omics in cancer

Second referee's report on "Integrated analysis of cervical squamous cell carcinoma cohorts from three continents reveals conserved subtypes of prognostic significance."

Background: In April 2020, I was part of a group of three refereed a very different version of this manuscript entitled "Pro-metastatic gene expression, immune evasion and an altered HPV spectrum characterize an aggressive subtype of cervical cancer" for Nature Cancer. We were reviewer 2. The manuscript was rejected and I think it is fair to judge that the concerns of reviewers 1 and 3 about the April 2020 submission were overlapping with but more serious than our concerns. Although 20 months have passed, the main conclusions of the new manuscript are different, the title is completely different, the author list is different, etc., the Associate Editor, Dr. McGinnis has asked me to treat this submission as a revised manuscript and to focus this report on the answer to the question: "Have your concerns been addressed in the revisions?"

My general answer is: In the rebuttal letter, the authors attempted to address our concerns. The manuscript is much improved. The writing is very good except for some figure legends that are too terse.

We thank the reviewer for taking the time to review our much-revised submission and we are glad to note that they consider it much improved. We hope that the further revisions we have made in light of their remaining concerns are satisfactory.

We have added detail to figure and supplementary figures throughout, examples are included in the response to reviewer #3's final comment (added text highlighted in red).

However, the authors failed to check carefully that everything they claimed to do in the rebuttal letter is actually done in the revised manuscript. I suspected this might be a problem because the rebuttal letter does not follow the widely used style that when a response/revision entails an insertion spanning one paragraph or less, then the inserted manuscript piece is quoted verbatim in the letter. Following below are two irrefutable examples from our comments and I believe there are other examples among the responses to reviewers 1 and 3:

We apologise for these instances, which unfortunately resulted from the scale of the revisions undertaken. We have check very carefully to ensure that we have now addressed all cases and have adopted the suggested practice of quoting verbatim, our changes to the text and signposting these with page and line numbers. We have also been through our responses to the original reviewer 1 and 3 comments and note in some instances for boxplots we had not defined the metrics represented by the boxes and whiskers as requested. We have also shown all datapoints on all boxplots, so the distributions of the data are as clear as possible. Examples of this are included in the response to reviewer #3's final comment.

In the rebuttal letter, the authors wrote: "We do however, acknowledge in our discussion, the need for more molecular profiling of cervical cancers from Africa, South America and Asia, where the vast majority of the cervical cancer burden falls." However, this issues is not discussed the revised Discussion.

We removed this statement after adding the CGCI data from Uganda during our extensive revisions but this valuable dataset notwithstanding, we agree that the need certainly remains. We've added the following statement to the discussion: "While we acknowledge there is an urgent need for more molecular data from parts of Africa, Asia and South America, where the vast majority of the disease burden now lies, it is notable that the key molecular and cellular characteristics of C1 and C2 tumours are consistent among cohorts included in this study from the USA, Europe, and Sub-Saharan Africa." Discussion, lines 513-517.

In our initial report, we suggested 10 references that should be cited because they provide(d) very relevant background information. In the rebuttal letter, the authors wrote gratefully that "We thank the reviewers for directing us to these key papers, several of which we have now cited in the revised submission and we apologise for the previous omissions." However, none of the suggested background papers are actually cited in the revised paper. I am not an author of any of these background papers, so my concern was and remains purely that the authors do not provide the background information as thoroughly as they could do to put their new findings in context.

We can only apologise for this, although we should like to point out that we did cite three of the suggested references (Chung *et al.*, 2019, Mayadav *et al.*, 2020 and Lheureux *et al.*, 2018), all of which are related to immune checkpoint blockade in cervical cancer. The point regarding background context is well-taken however and although we've moved away from HPV type comparisons in this revised manuscript (since we show that HPV type cannot explain the prognostic differences between clusters, nor is it an independent predictor of disease-specific survival in our validation cohort), we have now added the Guan *et al.*, (2012) and Sand *et al.* (2019) references to the citations for the following sentence in the introduction: "Cervical cancer is caused by at least 14 high-risk human

papillomaviruses (hrHPVs), with HPV16 and HPV18 together accounting for over 70% of cases worldwide, with some variation by region”. Introduction, lines 74-76.

Although the focus of this study has shifted from a comparison of alpha-9 versus alpha-7 HPV types, and therefore we’ve not referred to the studies on differences in the mechanisms by which HPV16 and HPV18 suppress host viral sensing and type 1 IFN responses, the general point regarding a lack of sufficient background is well-taken. To this end, in addition to the references mentioned above, we’ve added more background information on what is already known regarding the links between anti-tumour immune responses and prognosis in cervical cancer:

“The presence of tumour-reactive T-lymphocytes in the tumours and blood of cervical cancer patients are established prognostic factors, with the importance of reactivity against specific HPV peptides demonstrated for both cytotoxic and effector compartments, particularly in deeply-invasive tumours (Piersma *et al.* (2007); Heusinkfeld *et al.* (2011a); Gorter *et al.* (2015); Chen *et al.* (2021); Zhang *et al.* (2021); Cai *et al.* (2021))”. Results, lines 340-343.

And:

“IL-6 has also been implicated in the polarisation of macrophages towards a tolerogenic M2 state in cervical cancer (Heusinkfeld *et al.* (2011b))”. Discussion, lines 479-480.

We hope that the addition of this additional information (although necessarily brief due to word limits) and importantly, these citations, has helped to add balance to our manuscript and has better placed it in the context of our current knowledge of cervical cancer.

Overall, otherwise, I am generally satisfied that the authors’ new methods of analysis are rigorous. I am also satisfied that their new finding of two molecularly defined clusters of squamous cell cervical carcinoma with different overall survival is potentially important. The two clusters do not (yet) suggest different treatments but having information about different prognosis is still useful.

The following mostly minor items should be clarified or corrected:

1. In the legends of Figure 1b and Figure 3, it is not explained that the five rows above the main rectangle represent the cluster, the HPV type, the clade, stage, and grade. The color codes are confusing to me because, for example, my colorblind eyes perceive that the same shade of blue is being used to represent HPV16, Alpha 9 clade, Stage II, and G2 grade.

Thank you for pointing this out. We have changed the colour scheme for these figures and hope it is now clearer.

2. I do not understand the left part of Figure 4. Intuitively, how can it be that the ranking of lengths of the gray bars differs so much from the ranking by percent of tumors mutated on the right side. I understand that these two rankings will differ somewhat because the gray bars take into account the

gen length and the mutation frequencies do not consider the length, but the length alone cannot explain differences of a factor of 10 in gray bar lengths. I do not understand why the possible value of a gray bar is shown as 0.27, since $10^{-0.27} = 0.53$, which is not statistically significant. It seems that the left-hand side is described at line 622 as an “oncoplot” but without an explanation of how one is supposed to interpret an oncoplot.

dNdSCV does much more than simply accounting for differences in gene length when identifying significantly mutated genes, rather it calculates p and q values based on the odds that a gene is mutated more often than expected based on the background mutation rate computed using silent mutations while taking into account gene location, length, replication timing, substitution and base type, and genomic/nucleotide context. Genes that are highly recurrently mutated can still get low p/q values if the background model is deemed likely enough to have produced them. The objective behind such an analysis is to identify those somatic mutations that likely function as driver, rather than passenger events during tumour development. It is therefore not surprising that the model-based estimate of statistical significance diverges from the simple number/frequency of mutations when interpreting these plots.

Furthermore, in some cases (including the example discussed by the reviewer *DNAJB1*, where the q value = 0.53), a gene was found to be significantly mutated specifically among either C1 or C2 tumours but not among the entire cohort. These cases are annotated (‡) in the figure and shown in Supplementary Table S9.

We have added a much more thorough description of Figure 4 to the figure legend:

“Figure 4 Genomic summary of significantly mutated genes (SMGs) in SCC cohorts. The central plot shows mutation type and frequencies for 34 SMGs identified using dNdSCV on TCGA, Bergen and Ugandan cohorts (367 total patients). The histogram above the main plot represents tumour mutation burden (TMB, mutations / megabase) per sample. The histogram to the left displays the statistical significance of each SMG while the histogram to the right represents the proportion of C1 (blue) and C2 (red) tumours in which each gene is mutated. Black outlines around bars in this histogram signify a significant difference (FDR < 0.05, Wald Test) in mutation frequency between clusters. The stacked histogram below the main plot represents the breakdown of single base mutational signatures (SBS) detected in each tumour. Gene name key: blue = SMG in C1 tumours; red = SMG in C2 tumours; black = SMG in C1 and C2 tumours; black* = SMG only when combining C1 and C2 tumours; † = novel SMG in cervical cancer; ‡ = SMG in C1 tumours but not significant when C2 tumours also included.”

And some additional explanation in the Methods:

“...an oncoplot, which shows mutation rates and significance level for each SMG...” Methods, lines 686-687.

3. In Figure 5, I do not understand why the band 11q11 is shown in the two-digit representation and other bands, such as 6p22.1, are shown in the three-digit representation. It is not clear what were the

eligible CNA intervals for this analysis. When two tumors have overlapping CNAs with substantially different boundaries, it is unclear how it was decided whether these two events count as the same CNA or different CNAs. Amplifications adjacent bands 11q22.1 and 11q22.2 are depicted as distinct CNAs, but is it not the case that many tumors have a single amplification spanning both bands?

GISTIC2 amalgamates significant focal peaks if they are close enough, or if the peaks span two, whereas peaks that fall within a specific, more fine-grained cytoband and are extremely focal get designated with that. GISTIC2 tests for whether a wider peak has evidential support compared to a narrower peak too, which is how the significant peak boundaries are oft determined. Some tumours may indeed have CNAs that span multiple cytobands, but GISTIC2 defaults to the most focal peak that is statistically significant, and then nominates candidate driver genes that are subject to alterations at the core of the peak.

4. The Figure 6 legend fails to explain from what statistical test(s) the labels “ $q < 0.01$ ”, “false”, “true” are derived.

The statistical tests used for p and/or q-values and cutoffs used in each figure have now been added to the figure legends. For figure 6a it was Wilcoxon’s Rank Sum Test.

5. For many of the statistical tests (e.g., line 134, line 383), the authors do not indicate whether the test was one-sided or two-sided.

All tests are two sided by default, we have now added this information to the methods section:

“Statistical tests

Statistical tests were performed in R and in all cases were two-sided. Multiple test correction was performed using the `p.adjust` function in R base package and the Benjamini-Hochberg¹¹⁵ method where appropriate.” Methods, lines 767-770.

6. Since the authors observed that the gene *STK11* (lines 279 and 286) is a frequent target of somatic (and likely heterozygous) mutations in their cohort, they should comment on the incidence of cervical among patients with Peutz-Jeghers (cancer predisposition) syndrome in which patients typically have one allele of *STK11* mutated in the germline and the second allele mutated somatically in their tumors.

Thank you for drawing our attention to the link between PJS and cervical cancer. We have now added the following to the manuscript: “Finally, patients with Peutz-Jeghers Syndrome (PJS, an inherited tumour predisposition syndrome caused by germline *STK11* mutations) are at significantly increased risk of developing Minimal Deviation Adenocarcinoma, a rare and aggressive cervical malignancy that is not linked to HPV infection and sporadic cases of which frequently harbour somatic *STK11* mutations (Kuragaki *et al.*, 2003; Van Lier *et al.*, 2011; Wagner *et al.*, 2021).” Discussion, lines 466-471.

7. At lines 180-182, the difference in proportions between 20/25 and 39/69 is described as “interesting”, but the authors should report whether it is statistically significant by Fisher’s exact test.

The Fisher’s exact test p-value is 0.05. This has been added to the text at this point:

“Interestingly 80% (20/25) of Ugandan C2 patients were human immunodeficiency virus (HIV) positive, while only 56% (39/69) of C1 patients were HIV positive (OR = 0.33; p = 0.05, Fisher’s Exact Test, Supplementary Fig. S5a).” Results, lines 191-193.

8. In the references, the article titles and journal names are not formatted consistently. Some article titles have most words starting with UPPER CASE letters, while other titles have only the first word starting with an UPPER CASE letter. The same case inconsistency occurs for the journal titles that contain more than one word. Look for example at the consecutive references 97 (Genome biology) and 98 (Genome Biology). Why does reference 28 include the digital object identifier (doi)?

We’ve removed the doi’s from any references in which they were listed and have converted all journal names to title case (it seems this problem is a rather unhelpful quirk of Mendeley). The use of title case in the titles of articles however, carries through from how that was written by the authors themselves and unless the editor feels it necessary to convert these during production we respectfully suggest we leave the article titles as-is.

9. At a few places (e.g., lines 487, 518, 521, 522, 541, 561, etc.), the citations are not properly typeset as superscripts.

See response to point 8.

10. Line 667 should start “For cohorts for which RNASeq data were available...”

Done

“For cohorts for which RNAseq data were available (TCGA, Bergen and Uganda), a CAF index was calculated as described in Ko et al².” Methods 731-733.

11. Line 685 has an equals sign that was meant to be a double dash

Done

“positive cells; 4 = 201 – 300 positive cells; 5 = over 300 positive cells.” Methods, line 749.

12. The font color changed to non-black at lines 698-701.

Done

References

- Burk, R. D. *et al.* Integrated genomic and molecular characterization of cervical cancer. *Nature* **543**, 378–384 (2017).
- Cai, H. *et al.* HPV16 E6-specific T cell response and HLA-A alleles are related to the prognosis of patients with cervical cancer. *Infect. Agent. Cancer* **16**, 61 (2021).
- Ceccarelli, M. *et al.* Molecular Profiling Reveals Biologically Discrete Subsets and Pathways of Progression in Diffuse Glioma. *Cell* **164**, 550–563 (2016).
- Chen, Y., Li, Z.-Y., Zhou, G.-Q. & Sun, Y. An Immune-Related Gene Prognostic Index for Head and Neck Squamous Cell Carcinoma. *Clin. Cancer Res.* **27**, 330–341 (2021).
- Chumduri, C. *et al.* Opposing Wnt signals regulate cervical squamocolumnar homeostasis and emergence of metaplasia. *Nat. Cell Biol.* **23**, 184–197 (2021).
- Chung, H. C. *et al.* Efficacy and Safety of Pembrolizumab in Previously Treated Advanced Cervical Cancer: Results From the Phase II KEYNOTE-158 Study. *J. Clin. Oncol.* **37**, 1470–1478 (2019).
- Gagliardi, A. *et al.* Analysis of Ugandan cervical carcinomas identifies human papillomavirus clade-specific epigenome and transcriptome landscapes. *Nat. Genet.* **52**, 800–810 (2020).
- Galic, V. *et al.* Prognostic significance of adenocarcinoma histology in women with cervical cancer. *Gynecol. Oncol.* **125**, 287–291 (2012).
- Gilbert, D. C. *et al.* Tumour-infiltrating lymphocyte scores effectively stratify outcomes over and above p16 post chemo-radiotherapy in anal cancer. *Br. J. Cancer* **114**, 134–137 (2016).
- Gorter, A., Prins, F., van Diepen, M., Punt, S. & van der Burg, S. H. The tumor area occupied by Tbet+ cells in deeply invading cervical cancer predicts clinical outcome. *J. Transl. Med.* **13**, 295 (2015).
- Han, X. *et al.* Transdifferentiation of lung adenocarcinoma in mice with Lkb1 deficiency to squamous cell carcinoma. *Nat. Commun.* **5**, 3261 (2014).
- Heusinkveld, M. *et al.* The detection of circulating human papillomavirus-specific T cells is associated with improved survival of patients with deeply infiltrating tumors. *Int. J. Cancer* **128**, 379–389 (2011a).
- Heusinkveld, M. *et al.* M2 Macrophages Induced by Prostaglandin E₂ and IL-6 from Cervical Carcinoma Are Switched to Activated M1 Macrophages by CD4⁺ Th1 Cells. *J. Immunol.* **187**, 1157 (2011b).
- Hou, S., Han, X. & Ji, H. Squamous Transition of Lung Adenocarcinoma and Drug Resistance. *Trends in Cancer* **2**, 463–466 (2016).

Jung, E. J. *et al.* Cervical adenocarcinoma has a poorer prognosis and a higher propensity for distant recurrence than squamous cell carcinoma. *Int. J. Gynecol. Cancer* **27**, 1228–1236 (2017).

Kuragaki, C. *et al.* Mutations in the STK11 Gene Characterize Minimal Deviation Adenocarcinoma of the Uterine Cervix. *Lab. Investig.* **83**, 35–45 (2003).

Lheureux, S. *et al.* Association of Ipilimumab With Safety and Antitumor Activity in Women With Metastatic or Recurrent Human Papillomavirus-Related Cervical Carcinoma. *JAMA Oncol.* **4**, e173776–e173776 (2018).

Mayadev, J. S. *et al.* Sequential Ipilimumab After Chemoradiotherapy in Curative-Intent Treatment of Patients With Node-Positive Cervical Cancer. *JAMA Oncol.* **6**, 92–99 (2020).

Piersma, S. J. *et al.* High Number of Intraepithelial CD8+ Tumor-Infiltrating Lymphocytes Is Associated with the Absence of Lymph Node Metastases in Patients with Large Early-Stage Cervical Cancer. *Cancer Res.* **67**, 354–361 (2007).

Suissa, S. Immortal Time Bias in Pharmacoepidemiology. *Am. J. Epidemiol.* **167**, 492–499 (2008).

van Lier, M. G. F. *et al.* High cancer risk and increased mortality in patients with Peutz–Jeghers syndrome. *Gut* **60**, 141 (2011).

Wagner, A. *et al.* The Management of Peutz-Jeghers Syndrome: European Hereditary Tumour Group (EHTG) Guideline. *J. Clin. Med.* **10**, (2021).

Zhang, Y. *et al.* Baseline immunity and impact of chemotherapy on immune microenvironment in cervical cancer. *Br. J. Cancer* **124**, 414–424 (2021).

REVIEWER COMMENTS

Reviewer #1 (Remarks to the Author):

The authors present a revised manuscript, “Integrated analysis of cervical squamous cell carcinoma cohorts from three continents reveals conserved subtypes of prognostic significance”. I was a Reviewer for the version submitted to Nature Cancer in 2020. This manuscript is significantly tighter in data interpretation. In the letter of rebuttal, the authors have attempted to respond to our concerns. However, the authors have neglected to include the perspective of the practicing clinician. For example, although the authors

“explain in detail (Table 1 and first results section) how the overall cohort was broken down for the different analyses.”

All Stage I tumors are not the same. The authors should make this distinction explicitly. Treatment for Stage IA cancers is surgical. Treatment for Stage IB and above, depending on the exact clinical setting, begin to involve chemoradiation, or chemoradiation and surgery. The blanket designation of “Stage I” would raise an immediate red flag to anyone who takes care of cervical cancer patients.

The use of IHC to quantitate CD8 T-cells would be clarified simply by including the name of the system used, e.g. HALO. Examples of IHC images would also assist the reader in understanding why they were undertaken, and what was being analyzed. Something along the lines of “In a subset of cases, quantitative image analyses of the density of CD8+ cells were compared to data derived from the MethyCIBERSORT cell estimates from the same tissue blocks.”

The statement,

“The presence of tumor-reactive T-lymphocytes in the tumors and blood of cervical cancer patients are established prognostic factors, with the importance of reactivity against specific HPV peptides demonstrated for both cytotoxic and effector compartments, particularly in deeply-invasive tumors”

overstates the findings reported in references 33-41. It does not appear that any of the papers reports tumor-reactive T-lymphocytes in tumors.

Again referring to Table 1, survival by site is not so useful as survival by stage at each site. Survival by stage and C1/C2 would help to clarify the impact of this distinction.

Finally, in the end, what insights does this work provide that would inform future investigations, or could lead to changes in treatment strategies?

Regarding the utility of the distinction between C1 and C2, I defer to the Biostatistics Reviewer.

Reviewer #4 (Remarks to the Author):

Third review of

"Integrated 1 analysis of cervical squamous cell carcinoma cohorts from three continents reveals conserved subtypes of prognostic significance."

I reviewed earlier versions of this paper in April 2020 (for Nature Cancer, as part of a group of three) and again

in December 2021 (for Nature Communications). In each of these two instances, my review was the most favorable.

In this second revision, the authors have address my concerns with the first revision. However, I note that the concerns of the other reviewers were more serious.

I caught two problems with the revised Figure 1 or its legend:

Figure 1, it seems that the panel letters d and e got swapped.

In the forest plot, the number of individuals decreased from 236 to 229 without any explanation.

Reviewer #5 (Remarks to the Author):

Assessment of the degree to which the authors have responded to Reviewer #3's comments by Reviewer 3b (marked >), specialist in HPV genomics and evolution.

Reviewer #3 (Remarks to the Author): expertise in HPV virus evolution and pathology in cancer

This manuscript presents an analysis of cervix cancer TCGA multiomic data to evaluate 2 clusters they have identified for cancer outcome and survival. Improvements in classification of cancer that could distinguish different forms that might seem similar at the clinical or histologic level, but are different at the molecular level is of interest. Nevertheless, the manuscript has a number of shortcomings and deficiencies that need to be addressed.

We thank the reviewer for their careful critique of the student and welcome their positive comments relating to the utility of molecular classifiers for tumours that appear similar at the clinical or histologic level.

(1) Foremost, the description of the outcome – long term follow-up of cancer cases could use a more epidemiological approach. For instance, there is no mention of loss to follow-up, how was follow-up ascertained, etc. TCGA was initially designed to describe genetic changes in cancer, thus tumors that were larger in size would be more likely to be included, as would samples that could be surgically removed. This type of information needs to be in the manuscript and the authors should assess the exact starting cohorts, etc.

We fully agree with the reviewer on these points and apologise that this information was not presented more clearly in the previous version of the manuscript. Our primary motivation for curating a validation cohort in which to test inferences made from TCGA data was indeed the fact that for reasons including those highlighted by the reviewer, while TCGA is an invaluable resource for gaining molecular insight, making linkages to clinical outcome can be challenging. As detailed in our responses to Reviewer 1, we have now been able to present disease-specific survival for the combined European cohort, with only 9 of 313 patients lost to follow-up, which we feel strengthens the study considerably. We have also added detailed information on follow-up in a new Methods section (lines 548-565, copied below for reference). Table 1 contains an overview of the cohorts and Supplementary Table 6 contains detailed clinical and pathological data on a patient-by-patient basis, including detailed staging.

“Follow up of Oslo patients consisted of clinical examination every third month for the first two years, twice a year for the next three years, and once a year thereafter. When symptoms of relapse were detected, MRI of pelvis and retroperitoneum, and X-ray of thorax were performed. For the Innsbruck cohort, a total of 29 patients with invasive cervical squamous cell cancer (age 22-91 years; median 50 years), all treated between 1989 and 2010 at the Department of Obstetrics and Gynecology, Medical University of Innsbruck, were included in this study. The median observation time of all patients was 3.65 years (1331 days) (range = 0.06 – 21.54 years; 21 – 7,683 days) with respect to relapse-free survival. All patients were monitored in the outpatient follow-up program of the Department of Obstetrics and Gynecology, Medical University of Innsbruck. For the Bergen cohort, that were part of a prospective study, clinical and follow-up data for the Bergen cohort, which included patient age and diagnosis, clinical tumour size, FIGO stage and disease specific survival were collected by review of patient records and from correspondence with responsible gynaecologists if follow-up data were continued outside of the hospital. Further information on this

cohort can be found in the original study. Of the 313 total European cohort patients, nine were lost to follow up with a median of 13.7 years (5,012 days; range = 0.06 – 23.16 years; 21 – 8,455 days) to date last seen.”

>Comments reviewer 3b

Specifying the disease-specific survival of the combined European cohort with very little loss to follow up has adequately responded to Reviewer #3 suggestions. I am in agreement with the authors that this has strengthened the paper considerably. As to the Reviewer #3's comment regarding potential biases from tumour size and potential for surgical removal the authors refer to both the review undertaken in the Bergen cohort where tumor size and FIGO grade were recorded and to the original study (should be referenced here in the text following “Further information on this cohort can be found in the original study.”).

(2) The way the study was initially designed could lead to “immortal time bias” in the variables associated with survival.

According to Suissa (2008) “Immortal time refers to a span of time in the observation or follow-up period of a cohort during which the outcome under study could not have occurred. It usually occurs with the passing of time before a subject initiates a given exposure.”

Our study follows cohorts of patients who were only included if they underwent treatment for their cervical cancer (and thus there was a sample of their tumour available for molecular analysis. This occurred very rapidly following the date of diagnosis (used as T0 in all our survival analyses)). After this point, there is no time period during which the outcome under study (death due to any cause for TCGA or death due to cervical cancer for our validation cohort) could not have occurred and been recorded.

Chemo/radiotherapy is necessarily administered over a period following diagnosis (and surgery in those patients who received it). Any deaths that occurred during this time have been recorded however, therefore while we agree that a patient who died before completing a course of chemo/radiotherapy would not have received the entire treatment, this is an issue inherent in any such analysis of survival among cancer patients. We don't see how our study design could lead to immortal time bias but would be happy to explore this point further with the reviewer.

>Comment by reviewer 3b: I am in agreement with the authors that their study design does not inherently lead to immortal time bias.

(3) If this data is relevant to all cervical cancer, why exclude adenocarcinomas? This is a critical group to keep in the analyses, in fact they might demonstrate that in fact, cluster 2 is really adenocancers at the molecular level

We are not asserting that these data are relevant to all cervical cancer; rather we have presented a focused analysis of cervical squamous cell carcinoma, the most common histological subtype of cervical cancer. Numerous studies have shown clear differences in factors including prognosis, age at diagnosis, HPV type association and genomics between cervical adenocarcinoma and cervical SCC (e.g. Li et al., 2011; Jung et al., 2017; Galic et al., 2012; Burk et al., 2017). By restricting our analysis to SCC, we have avoided the risk of ‘identifying’ molecular associations that are simply a reflection of these well-documented differences in the biology and clinical behaviour of cervical adenocarcinoma and SCC. Many studies in cervical cancer and in other cancer types for which adenocarcinoma and SCC arise in the same tissue have similarly focused on one histologic subtype, notably in lung and oesophagus. Indeed TCGA conducted separate projects on lung adenocarcinoma (LUAD) and SCC (LUSC).

Although the SCCs in these cohorts are clearly distinct from adenocarcinomas at the whole transcriptome level (Supplementary Figure 1a-c), we do agree that cluster 2 shares certain molecular features with adenocarcinoma, as evidenced by enrichment of ‘columnar-like’ (Chumduri et al., 2021) tumours among C2 (Figure 3 and Discussion, lines 444-461). In our revised manuscript discussion (lines 461-466), “We note that evidence from mouse models has implicated Lkb1 (STK11) loss as a key event in enabling transition from lung adenocarcinoma to lung SCC; a transition that has been linked to drug resistance in lung cancer patients (Han et al., 2014; Hou et al., 2016). The frequent mutation and/or loss of STK11 in C2 tumours observed in our study suggests a similar adeno-SCC transition may be similarly linked to poor prognosis in cervical cancer.” Proving this speculation would require a complex set of lineage tracing experiments that is beyond the scope of the current study but we hope our observation will stimulate further research on this important topic. We verified that TCGA C2 tumours are indeed SCCs using the digital image archive (now added as Supplementary Table 12) and response to point (4) below).

>Comment by reviewer 3b: An interesting point raised by Reviewer #3. The authors however, make a clear case as to why they have focused their analysis to SCC only (adeno/squamous differences are well characterised). The interesting discovery made in the manuscript is that there are characterizable differences within the SCC assorting to two distinct clusters of prognostic value. The authors acknowledge that “cluster 2 shares certain molecular features with adenocarcinoma” and refers to a potential key transition event, the loss (or mutation) of STK11. This is very interesting and acknowledge the series of different events leading to SCC and the possible elusive cellular infection origin (now discussed). I agree with the authors in encouraging lineage tracing experiments in further research and that such investigations indeed are beyond the scope presented here.

(4) Given the HPV type differences in the cluster 1 and cluster 2 groups, a blinded analysis of the histology should be performed to determine if there was misclassification of cluster 2. As the author’s state, they seem to resemble glandular cancers and perhaps the histology was not so clear-cut.

Thank you for this suggestion – we had the same thought upon initial analysis of the molecular data. To address this, H&E-stained sections from TCGA’s digital image archive were reviewed by a histopathologist (Dr. J McDermott) who was blinded to TCGA’s histological designation

(Supplementary Table 12). In almost all cases, our pathologist was in agreement with TCGA's designation of these as CSCCs.

"This, taken together with the enrichment of alpha-7 HPV types in C2 tumours suggests that although they are SCCs (confirmed by examination of selected images from TCGA's digital image archive by a pathologist blinded to TCGA-assigned histology, Supplementary Table S12), they harbour features associated with adenocarcinoma; possibly even hinting at a different cell-of-origin for C1 versus C2 tumours". Discussion, lines 456-461.

>Comment by reviewer 3b :The authors have responded to Reviewer #3's suggestion by undertaking an independent histology review of a selection of 52 images (19 C2). From Table S12 one finds that the pathologist investigated a total of 34 TCGA-assigned SCC and 18 images TCGA-assigned adeno. Of the 34 TCGA assigned SCCs 13 were HPV16 and 18 HPV45; of the 18 TCGA assigned adenos, 15 were HPV16 and 3 HPV45. No histology of HPV18 positive tumours were investigated in either the SCC or the Adeno categories. The agreement however looks very good, and the authors write in their rebuttal that agreement was found in "almost all cases" which is somewhat unscientific. Notably, no TCGA assigned SCC was assigned adeno, but the adeno assignments seems more fuzzy in that two were assigned "poor(ly) diff SCC", and one "adenosquam". One could wish that also HPV18 positive tumours were reviewed when they outnumber the HPV45 infections (Table 1), although the two types are closely related (Alpha 7). One could also wish that more images overall were independently and blindly reviewed by more than one pathologist (standard practice), although the C2 SCCs are robust and coherent. Increasing the number could also compensate for those lost to poor quality, ambiguous assignments and missing images. Following this up would follow reviewer #3 concern that "perhaps the histology was not so clear cut" and improve the manuscript in convincing the reader that the SCCs are exactly defined as such by consensus independent of the TCGA entries. The finding of discrepancies would both educate the hint to different cell-of-origin (Discussion, lines 460-461) and the quality of the TCGA assignments. Agreements should be reported with a kappa-values.

Abstract – cervical cancer is a leading cause of cancer deaths in women.

We've updated the abstract accordingly (replaced "... represents one of the leading causes of cancer death worldwide" with "... is a leading cause of cancer deaths in women").

>Comment by reviewer 3b. OK

Line 81/82- HPV16 is also more common in adenocarcinoma, however the ratio of HPV16/HPV18 is vastly different.

Thank you for pointing out the ambiguity in this sentence. We've re-worded this sentence to quote the global statistics from 1990-2010 directly from the meta-analysis by Li et al (reference 2).

"HPV type is also associated with histology; HPV16 and HPV18 were reported in 59.3% and 13.2% of CSCC and in 36.3% and 36.8% of adenocarcinoma respectively worldwide, between 1990 and 2010²" Introduction, lines 81-84.

>Comment by reviewer 3b. OK

Table 1 needs footnotes. Percentages should be presented for stage, HPV type, HPV clade, etc.

We've included percentages and footnotes in our revised Table 1 as suggested.

>Comment by reviewer 3b. OK

Define TSNE.

We have included this in the first paragraph of the revised results section: "...were apparent in multi-dimensional t-distributed stochastic neighbour embedding (TSNE) analysis based on the top 10% most variable genes..." Results, lines 120-122.

>Comment by reviewer 3b. OK

Include statistical significance when reporting comparison of analyses, e.g., lines 180-182; 305-7, etc.

We have now added p-values and the statistical test used to derive them in all such comparisons, see examples below:

"Interestingly 80% (20/25) of Ugandan C2 patients were human immunodeficiency virus (HIV) positive, while only 56% (39/69) of C1 patients were HIV positive (OR = 0.33; $p = 0.05$, Fisher's Exact Test, Supplementary Fig. S5a)." Results, lines 191-193.

"There is considerable overlap between our C1 cluster and TCGA's 'mRNA-C2' cluster (84/106, odds ratio (OR) = 9, $p = 3.1 \times 10^{-7}$, Fisher's Exact Test) and keratin-high iCluster (80/106, OR = 44.7, $p = 9.8 \times 10^{-13}$, Fisher's Exact Test), and between our C2 cluster and TCGA's 'mRNA-C3' cluster (19/34, OR = 8.15, $p = 1.7 \times 10^{-6}$, Fisher's Exact Test) and keratin-low iCluster (27/34, OR = 18.3, $p = 5.02 \times 10^{-11}$, Fisher's Exact Test)." Results, lines 242-248.

>Comment by reviewer 3b. OK

Table 2 needs more information. What are the variables being analyzed? What was controlled for, etc.?

We've revised Table 2 to show that the comparisons are for C2 versus C1 and have included footnotes to indicate which covariates were included in each analysis. This information is also in the methods:

"Survival analyses of epigenetic allocations were carried out using Cox Proportional Hazards regression with age, tumour stage, HPV type, and with surgery, radiotherapy and chemotherapy (given/not given) as covariates" Methods, lines 645-648.

>Comment by reviewer 3b. ; Agree, OK

Figure legends need to describe all aspects of what is shown in the figures. For instance, there was no description of what rows represented in the heatmaps of figures 1-3.

We have included more detail throughout the figure (and supplementary figure) legends, as highlighted in red in the following examples:

"Figure 1 Derivation of prognostic clusters in TCGA SCC cohort. a) Consensus clustering of 236 TCGA HPV+ SCC patients produced an optimum 2 clusters (C1 and C2). Dark blue on the heat map represents patients (both row and column in heatmap) that clustered together 100% of the time while white represents patients that never clustered together after 90% sampling and replacement over 1000 iterations and considering only the 10% most variable genes. b) There were 938 differentially expressed genes between the two clusters. Expression values for each of the 938 genes is represented by the rows in the heatmap (columns represent individual patients). Kaplan-Meier curve shows that overall survival was higher in C1 patients in comparison to C2 patients when considering all patients (c). C2 is an individual predictor of prognosis, as is stage 4, while HPV type is not, as illustrated in a forest plot using cox-regression multivariate analysis using cluster type, age, stage and HPV type as covariates (d). Survival analysis using a Kaplan-Meier curve shows that overall survival was higher in C1 patients when considering only HPV16+ tumour patients (e). P-values on Kaplan-Meier curves from log-rank test, p-values in forest plot from Wald test."

"Figure 2 Cluster allocation of validation cohorts using methylation signature. A DNA methylation based signature of 129 methylation variable positions (MVPs; $dB > 0.25$, $FDR < 0.01$), represented by rows in heatmaps, separates clusters C1 and C2 patients in the TCGA cohort ($n = 236$) (a) and the validation dataset from the 3 European centres ($n = 313$) (b). c) Using DNA methylation data (BMIQ corrected beta values) for the TCGA and European cohorts, C2 tumours cluster together based on the 129 MVP signature. d) The disease specific survival (DSS) for C1 patients is higher than for C2 patients in the combined European validation cohort. (e) C2 is an individual predictor of prognosis,

as is stage 3 and 4, while HPV type is not, as illustrated in a forest plot using cox-regression multivariate analysis using cluster type, age, stage, HPV type and treatment regimen (RT = radiotherapy, CRT = chemoradiotherapy) as covariates. P-values on Kaplan-Meier curves from log-rank test, p-values in forest plot from Wald test.”

“Figure 3 Comparison of SCC subgroups with previous studies. Cluster analysis had previously been performed on 140 TCGA SCC tumours in two studies – one determined clusters based on cell of origin markers (Chumduri et al, 2021, red), and one determined clusters based on integrated omics data (TCGA Network, 2017, orange). Patients (columns) are identified as belonging to one of the clusters identified by Chumduri et al by red in the appropriate row and belonging to one of the TCGA clusters by orange in the appropriate row. C2 patients were more likely to belong to the Chumduri et al columnar-like cluster than the squamous-like cluster (odds ratio (OR) = 9.95, $p = 0.006$, Fisher’s Exact Test) and more likely to belong to the TCGA keratin-low iCluster (OR = 18.3, $p = 5.02 \times 10^{-11}$, Fisher’s Exact Test) than other iClusters. The heatmap at the bottom of plot represents expression levels of cytokeratin genes present in our C2 gene signature. 9 of the 14 genes exhibit low expression in C2 tumours, and 5 are highly expressed and so the TCGA “keratin-low” nomenclature was not used despite the large overlap with our C2 cluster.”

“Supplementary Figure S6 Elevation of epithelial mesenchymal transition (EMT) score is evident in C2 tumours. The EMT score derived by TCGA for 140 HPV+ squamous TCGA cervical cancer tumours is higher in the C2 compared to the C1 subgroup in our study. Each point represents a TCGA HPV+ tumour. The upper line in a box plot represents the upper quartile, the second line the median and the lowest line the lower quartile. The whisker above the box is drawn to the highest point within 1.5x the interquartile range (IQR), the whisker below the box is drawn to the lowest point within 1.5x the IQR. The P value is from Wilcoxon’s rank sum test.”

“Supplementary Figure S8 Increased levels of YAP in tumours with YAP1 amplification. YAP1 expression (a), and YAP protein levels (b) unphosphorylated and c) phosphorylated are higher in tumours that contain YAP1 amplifications. Blue points represent tumours without YAP1 amplification and red points represent tumours with YAP1 amplification. The upper line in a box plot represents the upper quartile, the second line the median and the lowest line the lower quartile. The whisker above the box is drawn to the highest point within 1.5x the interquartile range (IQR), the whisker below the box is drawn to the lowest point within 1.5x the IQR. P values are from Wilcoxon’s rank sum test.”

>Comment by reviewer 3b. Text is greatly improved and clear.

>Independent comment by reviewer 3b.

Line 755: Inconsistent writing “15min” whereas Line 757 “15 minutes”.

Point-by-point responses to reviewer comments:

Reviewer #1 (Remarks to the Author):

The authors present a revised manuscript, "Integrated analysis of cervical squamous cell carcinoma cohorts from three continents reveals conserved subtypes of prognostic significance". I was a Reviewer for the version submitted to Nature Cancer in 2020. This manuscript is significantly tighter in data interpretation. In the letter of rebuttal, the authors have attempted to respond to our concerns. However, the authors have neglected to include the perspective of the practicing clinician. For example, although the authors

"explain in detail (Table 1 and first results section) how the overall cohort was broken down for the different analyses."

All Stage I tumors are not the same. The authors should make this distinction explicitly. Treatment for Stage IA cancers is surgical. Treatment for Stage IB and above, depending on the exact clinical setting, begin to involve chemoradiation, or chemoradiation and surgery. The blanket designation of "Stage I" would raise an immediate red flag to anyone who takes care of cervical cancer patients.

Thank you for highlighting this. We now present detailed FIGO staging for every case (Table S1 for TCGA and Table S6 for the validation cohorts). We have also broken stage I cases into stage IA and IB in Table 1. Note that of the 643 cases in total, only 2 were stage IA. This is discussed in the revised manuscript (lines 225-228).

The use of IHC to quantitate CD8 T-cells would be clarified simply by including the name of the system used, e.g. HALO. Examples of IHC images would also assist the reader in understanding why they were undertaken, and what was being analyzed. Something along the lines of "In a subset of cases, quantitative image analyses of the density of CD8+ cells were compared to data derived from the MethylCIBERSORT cell estimates from the same tissue blocks."

For the Oslo samples that were stained for CD8 and for which digital quantification of staining was presented, an in-house platform developed in Matlab and described in Salberg UB et al BJC 2022 (DOI: 10.1038/s41416-022-01782-x) was used. We have added this brief description and citation to the Methods section (lines 792-794) and have included a Supplementary Methods document, in which we show example images along with detailed explanation. We hope this clarifies this point.

The statement,

"The presence of tumor-reactive T-lymphocytes in the tumors and blood of cervical cancer patients are established prognostic factors, with the importance of reactivity against specific HPV peptides demonstrated for both cytotoxic and effector compartments, particularly in deeply-invasive tumors"

overstates the findings reported in references 33-41. It does not appear that any of the papers reports tumor-reactive T-lymphocytes in tumors.

Thank you for bringing this to our attention. We have modified the sentence to more accurately reflect the conclusions of the cited studies as follows (lines 354-356):

"The presence of circulating HPV-reactive T-lymphocytes and of tumour-infiltrating cytotoxic T-lymphocytes have been associated with lower N-stage and improved prognosis in cervical cancer

patients..."

Again referring to Table 1, survival by site is not so useful as survival by stage at each site. Survival by stage and C1/C2 would help to clarify the impact of this distinction.

We have expanded Table S7, which is now entitled "Breakdown of patient numbers, vital status and survival for C1 and C2 patients by tumour stage". Where possible we have given 5-year survival rates (%) and have noted that for TCGA, this is OS and for the European cohort, this is DSS. Note that in all cases, except for the 14 TCGA patients with stage IV disease, survival rates by stage are higher for C1 than C2. This is discussed in the revised manuscript (lines 223-225).

Finally, in the end, what insights does this work provide that would inform future investigations, or could lead to changes in treatment strategies?

Key findings that will inform future investigations:

- 1) The identification of a subset of cervical SCCs (C2) which (although confirmed as SCCs by two independent pathologists in our team, see Table S12), display gene expression profiles, a prevalence of alpha-7 HPV types and genomic alterations that are characteristic of adenocarcinoma. This suggests a possible difference in cell-of-origin and/or transition event (see point 2), in this SCC subset that will be of considerable interest for further investigation (for example by using lineage tracing in mouse models as we suggest in the manuscript).
- 2) The identification of *STK11* loss and/or mutation as a frequent event in C2 tumours that may (as has been observed in lung cancer), drive a transition from adenocarcinoma to SCC.
- 3) The identification of potential targets for targeted therapies in this poor prognosis subgroup, including YAP1 and the immune checkpoint proteins NT5E, B7-H3 and PD-L2.
- 4) The identification of 26 significantly mutated genes not previously implicated in cervical SCC will inform functional studies on these genes and their role in cervical cancer pathogenesis, potentially enabling identification of further potential therapeutic targets.
- 5) Although larger numbers are needed for robust within-stage comparisons of C1 and C2 tumours, we observe a clear trend in the survival rates between C1 and C2 by stage (data now added to Table S7). Taking molecular (C1/C2) subtyping into account may therefore allow for more accurate prognostication than current staging and potentially (clearly dependent upon prospective studies) different clinical management of patients with C1 versus C2 tumours.

We feel we have touched upon all the above points in the manuscript without overstating the significance of our findings but we are happy to modify the Discussion and Abstract further if the reviewer and/or editors deem it necessary.

Regarding the utility of the distinction between C1 and C2, I defer to the Biostatistics Reviewer.

Reviewer #4 (Remarks to the Author):

Third review of

"Integrated 1 analysis of cervical squamous cell carcinoma cohorts from three continents reveals conserved subtypes of prognostic significance."

I reviewed earlier versions of this paper in April 2020 (for Nature Cancer, as part of a group of three) and again

in December 2021 (for Nature Communications). In each of these two instances, my review was the most favorable.

In this second revision, the authors have address my concerns with the first revision. However, I note that the concerns of the other reviewers were more serious.

I caught two problems with the revised Figure 1 or its legend:

Figure 1, it seems that the panel letters d and e got swapped.

Thank you for drawing this to our attention. To fit the figure panels into the space available while retaining clarity, we placed Figure 1D below 1E but we have modified the legend with the aim of making this clearer. If deemed necessary by the editor, we are happy to re-position these figure panels in the final version for production.

In the forest plot, the number of individuals decreased from 236 to 229 without any explanation.

We included this explanation in the Results section on the Cox regression (lines 147-148 in this version): “n=229 patients for whom data on all covariates were complete”) but we have also now added this information to the figure legend (manuscript lines 1152-1153).

Reviewer #5 (Remarks to the Author):

Assessment of the degree to which the authors have responded to Reviewer #3's comments by Reviewer 3b (marked >), specialist in HPV genomics and evolution.

Reviewer #3 (Remarks to the Author): expertise in HPV virus evolution and pathology in cancer

This manuscript presents an analysis of cervix cancer TCGA multiomic data to evaluate 2 clusters they have identified for cancer outcome and survival. Improvements in classification of cancer that could distinguish different forms that might seem similar at the clinical or histologic level, but are different at the molecular level is of interest. Nevertheless, the manuscript has a number of shortcomings and deficiencies that need to be addressed.

We thank the reviewer for their careful critique of the student and welcome their positive comments relating to the utility of molecular classifiers for tumours that appear similar at the clinical or histologic level.

(1) Foremost, the description of the outcome – long term follow-up of cancer cases could use a more epidemiological approach. For instance, there is no mention of loss to follow-up, how was follow-up ascertained, etc. TCGA was initially designed to describe genetic changes in cancer, thus tumors that were larger in size would be more likely to be included, as would samples that could be surgically removed. This type of information needs to be in the manuscript and the authors should assess the exact starting cohorts, etc.

We fully agree with the reviewer on these points and apologise that this information was not presented more clearly in the previous version of the manuscript. Our primary motivation for curating a validation cohort in which to test inferences made from TCGA data was indeed the fact that for reasons including those highlighted by the reviewer, while TCGA is an invaluable resource for gaining molecular insight, making linkages to clinical outcome can be challenging. As detailed in

our responses to Reviewer 1, we have now been able to present disease-specific survival for the combined European cohort, with only 9 of 313 patients lost to follow-up, which we feel strengthens the study considerably. We have also added detailed information on follow-up in a new Methods section (lines 548-565, copied below for reference). Table 1 contains an overview of the cohorts and Supplementary Table 6 contains detailed clinical and pathological data on a patient-by-patient basis, including detailed staging.

“Follow up of Oslo patients consisted of clinical examination every third month for the first two years, twice a year for the next three years, and once a year thereafter. When symptoms of relapse were detected, MRI of pelvis and retroperitoneum, and X-ray of thorax were performed. For the Innsbruck cohort, a total of 29 patients with invasive cervical squamous cell cancer (age 22-91 years; median 50 years), all treated between 1989 and 2010 at the Department of Obstetrics and Gynecology, Medical University of Innsbruck, were included in this study. The median observation time of all patients was 3.65 years (1331 days) (range = 0.06 – 21.54 years; 21 – 7,683 days) with respect to relapse-free survival. All patients were monitored in the outpatient follow-up program of the Department of Obstetrics and Gynecology, Medical University of Innsbruck. For the Bergen cohort, that were part of a prospective study, clinical and follow-up data for the Bergen cohort, which included patient age and diagnosis, clinical tumour size, FIGO stage and disease specific survival were collected by review of patient records and from correspondence with responsible gynaecologists if follow-up data were continued outside of the hospital. Further information on this cohort can be found in the original study. Of the 313 total European cohort patients, nine were lost to follow up with a median of 13.7 years (5,012 days; range = 0.06 – 23.16 years; 21 – 8,455 days) to date last seen.”

>Comments reviewer 3b

Specifying the disease-specific survival of the combined European cohort with very little loss to follow up has adequately responded to Reviewer #3 suggestions. I am in agreement with the authors that this has strengthened the paper considerably. As to the Reviewer #3's comment regarding potential biases from tumour size and potential for surgical removal the authors refer to both the review undertaken in the Bergen cohort where tumor size and FIGO grade were recorded and to the original study (should be referenced here in the text following “Further information on this cohort can be found in the original study.”).

(2) The way the study was initially designed could lead to “immortal time bias” in the variables associated with survival.

According to Suissa (2008) “Immortal time refers to a span of time in the observation or follow-up period of a cohort during which the outcome under study could not have occurred. It usually occurs with the passing of time before a subject initiates a given exposure.”

Our study follows cohorts of patients who were only included if they underwent treatment for their cervical cancer (and thus there was a sample of their tumour available for molecular analysis. This occurred very rapidly following the date of diagnosis (used as T0 in all our survival analyses)). After this point, there is no time period during which the outcome under study (death due to any cause for TCGA or death due to cervical cancer for our validation cohort) could not have occurred and been recorded.

Chemo/radiotherapy is necessarily administered over a period following diagnosis (and surgery in those patients who received it). Any deaths that occurred during this time have been recorded however, therefore while we agree that a patient who died before completing a course of chemo/radiotherapy would not have received the entire treatment, this is an issue inherent in any such analysis of survival among cancer patients. We don't see how our study design could lead to

immortal time bias but would be happy to explore this point further with the reviewer.

>Comment by reviewer 3b: I am in agreement with the authors that their study design does not inherently lead to immortal time bias.

(3) If this data is relevant to all cervical cancer, why exclude adenocarcinomas? This is a critical group to keep in the analyses, in fact they might demonstrate that in fact, cluster 2 is really adenocancers at the molecular level

We are not asserting that these data are relevant to all cervical cancer; rather we have presented a focused analysis of cervical squamous cell carcinoma, the most common histological subtype of cervical cancer. Numerous studies have shown clear differences in factors including prognosis, age at diagnosis, HPV type association and genomics between cervical adenocarcinoma and cervical SCC (e.g. Li et al., 2011; Jung et al., 2017; Galic et al., 2012; Burk et al., 2017). By restricting our analysis to SCC, we have avoided the risk of ‘identifying’ molecular associations that are simply a reflection of these well-documented differences in the biology and clinical behaviour of cervical adenocarcinoma and SCC. Many studies in cervical cancer and in other cancer types for which adenocarcinoma and SCC arise in the same tissue have similarly focused on one histologic subtype, notably in lung and oesophagus. Indeed TCGA conducted separate projects on lung adenocarcinoma (LUAD) and SCC (LUSC).

Although the SCCs in these cohorts are clearly distinct from adenocarcinomas at the whole transcriptome level (Supplementary Figure 1a-c), we do agree that cluster 2 shares certain molecular features with adenocarcinoma, as evidenced by enrichment of ‘columnar-like’ (Chumduri et al., 2021) tumours among C2 (Figure 3 and Discussion, lines 444-461). In our revised manuscript discussion (lines 461-466), “We note that evidence from mouse models has implicated Lkb1 (STK11) loss as a key event in enabling transition from lung adenocarcinoma to lung SCC; a transition that has been linked to drug resistance in lung cancer patients (Han et al., 2014; Hou et al., 2016). The frequent mutation and/or loss of STK11 in C2 tumours observed in our study suggests a similar adeno-SCC transition may be similarly linked to poor prognosis in cervical cancer.” Proving this speculation would require a complex set of lineage tracing experiments that is beyond the scope of the current study but we hope our observation will stimulate further research on this important topic. We verified that TCGA C2 tumours are indeed SCCs using the digital image archive (now added as Supplementary Table 12) and response to point (4) below).

>Comment by reviewer 3b: An interesting point raised by Reviewer #3. The authors however, make a clear case as to why they have focused their analysis to SCC only (adeno/squamous differences are well characterised). The interesting discovery made in the manuscript is that there are characterizable differences within the SCC assorting to two distinct clusters of prognostic value. The authors acknowledge that “cluster 2 shares certain molecular features with adenocarcinoma” and refers to a potential key transition event, the loss (or mutation) of STK11. This is very interesting and acknowledge the series of different events leading to SCC and the possible elusive cellular infection origin (now discussed). I agree with the authors in encouraging lineage tracing experiments in further research and that such investigations indeed are beyond the scope presented here.

(4) Given the HPV type differences in the cluster 1 and cluster 2 groups, a blinded analysis of the histology should be performed to determine if there was misclassification of cluster 2. As the author’s state, they seem to resemble glandular cancers and perhaps the histology was not so clear-cut.

Thank you for this suggestion – we had the same thought upon initial analysis of the molecular data.

To address this, H&E-stained sections from TCGA's digital image archive were reviewed by a histopathologist (Dr. J McDermott) who was blinded to TCGA's histological designation (Supplementary Table 12). In almost all cases, our pathologist was in agreement with TCGA's designation of these as CSCCs.

"This, taken together with the enrichment of alpha-7 HPV types in C2 tumours suggests that although they are SCCs (confirmed by examination of selected images from TCGA's digital image archive by a pathologist blinded to TCGA-assigned histology, Supplementary Table S12), they harbour features associated with adenocarcinoma; possibly even hinting at a different cell-of-origin for C1 versus C2 tumours". Discussion, lines 456-461.

>Comment by reviewer 3b :The authors have responded to Reviewer #3's suggestion by undertaking an independent histology review of a selection of 52 images (19 C2). From Table S12 one finds that the pathologist investigated a total of 34 TCGA-assigned SCC and 18 images TCGA-assigned adeno. Of the 34 TCGA assigned SCCs 13 were HPV16 and 18 HPV45; of the 18 TCGA assigned adenos, 15 were HPV16 and 3 HPV45. No histology of HPV18 positive tumours were investigated in either the SCC or the Adeno categories. The agreement however looks very good, and the authors write in their rebuttal that agreement was found in "almost all cases" which is somewhat unscientific. Notably, no TCGA assigned SCC was assigned adeno, but the adeno assignments seems more fuzzy in that two were assigned "poor(ly) diff SCC", and one "adenosquam". One could wish that also HPV18 positive tumours were reviewed when they outnumber the HPV45 infections (Table 1), although the two types are closely related (Alpha 7). One could also wish that more images overall were independently and blindly reviewed by more than one pathologist (standard practice), although the C2 SCCs are robust and coherent. Increasing the number could also compensate for those lost to poor quality, ambiguous assignments and missing images. Following this up would follow reviewer #3 concern that "perhaps the histology was not so clear cut" and improve the manuscript in convincing the reader that the SCCs are exactly defined as such by consensus independent of the TCGA entries. The finding of discrepancies would both educate the hint to different cell-of-origin (Discussion, lines 460-461) and the quality of the TCGA assignments. Agreements should be reported with a kappa-values.

Thank you for these suggestions. We have now added review of the histology for all HPV18+ SCC cases for which evaluable images were available. Independent review of a total of 75 TCGA cases has now been undertaken by two pathologists (Dr J. McDermott and Dr G. Thomas) with close agreement (Cohen's Kappa = 0.89, 95% CI 0.77 – 1). This information has been added to the Results section (lines 475-478) and to Table S12. We've added a statement to this effect also to the Methods (lines 684-687).

Abstract – cervical cancer is a leading cause of cancer deaths in women.

We've updated the abstract accordingly (replaced "... represents one of the leading causes of cancer death worldwide" with "... is a leading cause of cancer deaths in women").

>Comment by reviewer 3b. OK

Line 81/82- HPV16 is also more common in adenocarcinoma, however the ratio of HPV16/HPV18 is vastly different.

Thank you for pointing out the ambiguity in this sentence. We've re-worded this sentence to quote the global statistics from 1990-2010 directly from the meta-analysis by Li et al (reference 2). "HPV type is also associated with histology; HPV16 and HPV18 were reported in 59.3% and 13.2% of CSCC and in 36.3% and 36.8% of adenocarcinoma respectively worldwide, between 1990 and 2010"

Introduction, lines 81-84.

>Comment by reviewer 3b. OK

Table 1 needs footnotes. Percentages should be presented for stage, HPV type, HPV clade, etc.

We've included percentages and footnotes in our revised Table 1 as suggested.

>Comment by reviewer 3b. OK

Define TSNE.

We have included this in the first paragraph of the revised results section: "...were apparent in multi-dimensional t-distributed stochastic neighbour embedding (TSNE) analysis based on the top 10% most variable genes..." Results, lines 120-122.

>Comment by reviewer 3b. OK

Include statistical significance when reporting comparison of analyses, e.g., lines 180-182; 305-7, etc.

We have now added p-values and the statistical test used to derive them in all such comparisons, see examples below:

"Interestingly 80% (20/25) of Ugandan C2 patients were human immunodeficiency virus (HIV) positive, while only 56% (39/69) of C1 patients were HIV positive (OR = 0.33; $p = 0.05$, Fisher's Exact Test, Supplementary Fig. S5a)." Results, lines 191-193.

"There is considerable overlap between our C1 cluster and TCGA's 'mRNA-C2' cluster (84/106, odds ratio (OR) = 9, $p = 3.1 \times 10^{-7}$, Fisher's Exact Test) and keratin-high iCluster (80/106, OR = 44.7, $p = 9.8 \times 10^{-13}$, Fisher's Exact Test), and between our C2 cluster and TCGA's 'mRNA-C3' cluster (19/34, OR = 8.15, $p = 1.7 \times 10^{-6}$, Fisher's Exact Test) and keratin-low iCluster (27/34, OR = 18.3, $p = 5.02 \times 10^{-11}$, Fisher's Exact Test)." Results, lines 242-248.

>Comment by reviewer 3b. OK

Table 2 needs more information. What are the variables being analyzed? What was controlled for, etc.?

We've revised Table 2 to show that the comparisons are for C2 versus C1 and have included footnotes to indicate which covariates were included in each analysis. This information is also in the methods:

"Survival analyses of epigenetic allocations were carried out using Cox Proportional Hazards regression with age, tumour stage, HPV type, and with surgery, radiotherapy and chemotherapy (given/not given) as covariates" Methods, lines 645-648.

>Comment by reviewer 3b. ; Agree, OK

Figure legends need to describe all aspects of what is shown in the figures. For instance, there was no description of what rows represented in the heatmaps of figures 1-3.

We have included more detail throughout the figure (and supplementary figure) legends, as

highlighted in red in the following examples:

“Figure 1 Derivation of prognostic clusters in TCGA SCC cohort. a) Consensus clustering of 236 TCGA HPV+ SCC patients produced an optimum 2 clusters (C1 and C2). Dark blue on the heat map represents patients (both row and column in heatmap) that clustered together 100% of the time while white represents patients that never clustered together after 90% sampling and replacement over 1000 iterations and considering only the 10% most variable genes. b) There were 938 differentially expressed genes between the two clusters. Expression values for each of the 938 genes is represented by the rows in the heatmap (columns represent individual patients). Kaplan-Meier curve shows that overall survival was higher in C1 patients in comparison to C2 patients when considering all patients (c). C2 is an individual predictor of prognosis, as is stage 4, while HPV type is not, as illustrated in a forest plot using cox-regression multivariate analysis using cluster type, age, stage and HPV type as covariates (d). Survival analysis using a Kaplan-Meier curve shows that overall survival was higher in C1 patients when considering only HPV16+ tumour patients (e). P-values on Kaplan-Meier curves from log-rank test, p-values in forest plot from Wald test.”

“Figure 2 Cluster allocation of validation cohorts using methylation signature. A DNA methylation based signature of 129 methylation variable positions (MVPs; $dB > 0.25$, $FDR < 0.01$), represented by rows in heatmaps, separates clusters C1 and C2 patients in the TCGA cohort ($n = 236$) (a) and the validation dataset from the 3 European centres ($n = 313$) (b). c) Using DNA methylation data (BMIQ corrected beta values) for the TCGA and European cohorts, C2 tumours cluster together based on the 129 MVP signature. d) The disease specific survival (DSS) for C1 patients is higher than for C2 patients in the combined European validation cohort. (e) C2 is an individual predictor of prognosis, as is stage 3 and 4, while HPV type is not, as illustrated in a forest plot using cox-regression multivariate analysis using cluster type, age, stage, HPV type and treatment regimen (RT = radiotherapy, CRT = chemoradiotherapy) as covariates. P-values on Kaplan-Meier curves from log-rank test, p-values in forest plot from Wald test.”

“Figure 3 Comparison of SCC subgroups with previous studies. Cluster analysis had previously been performed on 140 TCGA SCC tumours in two studies – one determined clusters based on cell of origin markers (Chumduri et al, 2021, red), and one determined clusters based on integrated omics data (TCGA Network, 2017, orange). Patients (columns) are identified as belonging to one of the clusters identified by Chumduri et al by red in the appropriate row and belonging to one of the TCGA clusters by orange in the appropriate row. C2 patients were more likely to belong to the Chumduri et al columnar-like cluster than the squamous-like cluster (odds ratio (OR) = 9.95, $p = 0.006$, Fisher’s Exact Test) and more likely to belong to the TCGA keratin-low iCluster (OR = 18.3, $p = 5.02 \times 10^{-11}$, Fisher’s Exact Test) than other iClusters. The heatmap at the bottom of plot represents expression levels of cytokeratin genes present in our C2 gene signature. 9 of the 14 genes exhibit low expression in C2 tumours, and 5 are highly expressed and so the TCGA “keratin-low” nomenclature was not used despite the large overlap with our C2 cluster.”

“Supplementary Figure S6 Elevation of epithelial mesenchymal transition (EMT) score is evident in C2 tumours. The EMT score derived by TCGA for 140 HPV+ squamous TCGA cervical cancer tumours is higher in the C2 compared to the C1 subgroup in our study. Each point represents a TCGA HPV+ tumour. The upper line in a box plot represents the upper quartile, the second line the median and the lowest line the lower quartile. The whisker above the box is drawn to the highest point within 1.5x the interquartile range (IQR), the whisker below the box is drawn to the lowest point within 1.5x the IQR. The P value is from Wilcoxon’s rank sum test.”

“Supplementary Figure S8 Increased levels of YAP in tumours with YAP1 amplification. YAP1

expression (a), and YAP protein levels (b) unphosphorylated and c) phosphorylated are higher in tumours that contain YAP1 amplifications. Blue points represent tumours without YAP1 amplification and red points represent tumours with YAP1 amplification. The upper line in a box plot represents the upper quartile, the second line the median and the lowest line the lower quartile. The whisker above the box is drawn to the highest point within 1.5x the interquartile range (IQR), the whisker below the box is drawn to the lowest point within 1.5x the IQR. P values are from Wilcoxon's rank sum test."

>Comment by reviewer 3b. Text is greatly improved and clear.

>Independent comment by reviewer 3b.

Line 755: Inconsistent writing "15min" whereas Line 757 "15 minutes".

Fixed.

REVIEWERS' COMMENTS

Reviewer #6 (Remarks to the Author):

Given my expertise in cervical clinical oncology and tumor microenvironment, I have been asked to assess the adequacy of response of the authors to reviewer 1. First and foremost, I want to thank the authors for conducting this important study and for their responsiveness to the multiple questions raised by the number of reviewers. My assessment will be marked as Reviewer 1b.

Reviewer 1

The authors present a revised manuscript, "Integrated analysis of cervical squamous cell carcinoma cohorts from three continents reveals conserved subtypes of prognostic significance". I was a Reviewer for the version submitted to Nature Cancer in 2020. This manuscript is significantly tighter in data interpretation. In the letter of rebuttal, the authors have attempted to respond to our concerns. However, the authors have neglected to include the perspective of the practicing clinician. For example, although the authors "explain in detail (Table 1 and first results section) how the overall cohort was broken down for the different analyses." All Stage I tumors are not the same. The authors should make this distinction explicitly. Treatment for Stage IA cancers is surgical. Treatment for Stage IB and above, depending on the exact clinical setting, begin to involve chemoradiation, or chemoradiation and surgery. The blanket designation of "Stage I" would raise an immediate red flag to anyone who takes care of cervical cancer patients.

Author response:

Thank you for highlighting this. We now present detailed FIGO staging for every case (Table S1 for TCGA and Table S6 for the validation cohorts). We have also broken stage I cases into stage IA and IB in Table 1. Note that of the 643 cases in total, only 2 were stage IA. This is discussed in the revised manuscript (lines 225-228).

Comment by reviewer 1b: Thank you for this detailed response. Given that the samples were collected over a number of years, and the FIGO staging system has evolved over time, the authors need to indicate (perhaps in Methods) the FIGO staging criteria that were used in each cohort (e.g. 2009, 2018).

Reviewer 1

The use of IHC to quantitate CD8 T-cells would be clarified simply by including the name of the system used, e.g. HALO. Examples of IHC images would also assist the reader in understanding why they were undertaken, and what was being analyzed. Something along the lines of "In a subset of cases, quantitative image analyses of the density of CD8+ cells were compared to data derived from the MethyCIBERSORT cell estimates from the same tissue blocks."

Author response

For the Oslo samples that were stained for CD8 and for which digital quantification of staining was presented, an in-house platform developed in Matlab and described in Salberg UB et al BJC 2022 (DOI: 10.1038/s41416-022-01782-x) was used. We have added this brief description and citation to the Methods section (lines 792-794) and have included a Supplementary Methods document, in which we show example images along with detailed explanation. We hope this clarifies this point.

Comment by reviewer 1b: Thank you for the clarification and for inclusion of the methods.

Reviewer 1

The statement, "The presence of tumor-reactive T-lymphocytes in the tumors and blood of cervical cancer patients are established prognostic factors, with the importance of reactivity against specific HPV peptides demonstrated for both cytotoxic and effector compartments, particularly in deeply-invasive tumors" overstates the findings reported in references 33-41. It does not appear that any of the papers reports tumor-reactive T-lymphocytes in tumors.

Author response

Thank you for bringing this to our attention. We have modified the sentence to more accurately reflect the conclusions of the cited studies as follows (lines 354-356):
"The presence of circulating HPV-reactive T-lymphocytes and of tumour-infiltrating cytotoxic T lymphocytes have been associated with lower N-stage and improved prognosis in cervical cancer patients..."

Comment by reviewer 1b: Ok.

Reviewer 1

Again referring to Table 1, survival by site is not so useful as survival by stage at each site. Survival by stage and C1/C2 would help to clarify the impact of this distinction.

Author response

We have expanded Table S7, which is now entitled "Breakdown of patient numbers, vital status and survival for C1 and C2 patients by tumour stage". Where possible we have given 5-year survival rates (%) and have noted that for TCGA, this is OS and for the European cohort, this is DSS. Note that in all cases, except for the 14 TCGA patients with stage IV disease, survival rates by stage are higher for C1 than C2. This is discussed in the revised manuscript (lines 223-225).

Comment by reviewer 1b: Thank you for providing the by-stage analyses. Even without robust statistics, the data nevertheless support the fact that the findings are not driven by different stage overrepresentation in C1 vs. C2.

Reviewer 1

Finally, in the end, what insights does this work provide that would inform future investigations, or could lead to changes in treatment strategies?

Author response

Key findings that will inform future investigations:

- 1) The identification of a subset of cervical SCCs (C2) which (although confirmed as SCCs by two independent pathologists in our team, see Table S12), display gene expression profiles, a prevalence of alpha-7 HPV types and genomic alterations that are characteristic of adenocarcinoma. This suggests a possible difference in cell-of-origin and/or transition event (see point 2), in this SCC subset that will be of considerable interest for further investigation (for example by using lineage tracing in mouse models as we suggest in the manuscript).
- 2) The identification of STK11 loss and/or mutation as a frequent event in C2 tumours that may (as has been observed in lung cancer), drive a transition from adenocarcinoma to SCC.
- 3) The identification of potential targets for targeted therapies in this poor prognosis subgroup, including YAP1 and the immune checkpoint proteins NT5E, B7-H3 and PD-L2.
- 4) The identification of 26 significantly mutated genes not previously implicated in cervical SCC will inform functional studies on these genes and their role in cervical cancer pathogenesis, potentially enabling identification of further potential therapeutic targets.
- 5) Although larger numbers are needed for robust within-stage comparisons of C1 and C2 tumours, we observe a clear trend in the survival rates between C1 and C2 by stage (data now added to Table S7). Taking molecular (C1/C2) subtyping into account may therefore allow for more accurate prognostication than current staging and potentially (clearly dependent upon prospective studies) different clinical management of patients with C1 versus C2 tumours. We feel we have touched upon all the above points in the manuscript without overstating the significance of our findings but we are happy to modify the Discussion and Abstract further if the reviewer and/or editors deem it necessary.

Comment by reviewer 1b: The points above are valid, but are not well-elaborated in the discussion and I think the investigators could provide additional clinical implications of the results. For example, aside from the prognostic significance, C1 vs. C2 distinction may identify patient populations at risk for relapse that require further adjuvant therapy after completion of upfront therapy. There are also implications for exploring C1 vs. C2 as a predictor of response within the

context of approved agents (e.g. pembrolizumab), but also within the context of ongoing clinical trials with other agents.

RESPONSE TO REVIEWERS' COMMENTS

Reviewer #6 (Remarks to the Author):

Given my expertise in cervical clinical oncology and tumor microenvironment, I have been asked to assess the adequacy of response of the authors to reviewer 1. First and foremost, I want to thank the authors for conducting this important study and for their responsiveness to the multiple questions raised by the number of reviewers. My assessment will be marked as Reviewer 1b.

We thank the reviewer for their kind words about our study and for their willingness to evaluate our revisions.

Reviewer 1

The authors present a revised manuscript, "Integrated analysis of cervical squamous cell carcinoma cohorts from three continents reveals conserved subtypes of prognostic significance". I was a Reviewer for the version submitted to Nature Cancer in 2020. This manuscript is significantly tighter in data interpretation. In the letter of rebuttal, the authors have attempted to respond to our concerns. However, the authors have neglected to include the perspective of the practicing clinician. For example, although the authors "explain in detail (Table 1 and first results section) how the overall cohort was broken down for the different analyses." All Stage I tumors are not the same. The authors should make this distinction explicitly. Treatment for Stage IA cancers is surgical. Treatment for Stage IB and above, depending on the exact clinical setting, begin to involve chemoradiation, or chemoradiation and surgery. The blanket designation of "Stage I" would raise an immediate red flag to anyone who takes care of cervical cancer patients.

Author response:

Thank you for highlighting this. We now present detailed FIGO staging for every case (Table S1 for TCGA and Table S6 for the validation cohorts). We have also broken stage I cases into stage IA and IB in Table 1. Note that of the 643 cases in total, only 2 were stage IA. This is discussed in the revised manuscript (lines 225-228).

Comment by reviewer 1b: Thank you for this detailed response. Given that the samples were collected over a number of years, and the FIGO staging system has evolved over time, the authors need to indicate (perhaps in Methods) the FIGO staging criteria that were used in each cohort (e.g. 2009, 2018).

We have added this information to the Methods where it was possible to ascertain. The 2009 FIGO staging system was used for the Oslo (n=248), Bergen (n=37) and Uganda cohorts (n=94) and for the Innsbruck cohort (n=28), in which samples originate from the period 1989-2010, clinical classification corresponds to the classification valid at that time. Unfortunately, we were unable to ascertain the version of the FIGO classification used for the TCGA cohort (n=236) but it is certainly pre-2018.

We have added the following sentence to the 'Patient Samples' section of the Methods: "Tumours in the Oslo and Bergen cohorts were staged according to the 2009 FIGO staging system for cervical cancer and tumours in the Innsbruck cohort were staged according to the FIGO staging system valid at the time of diagnosis (1989-2010)."

This information on staging of the Uganda cohort is included in the paper we reference for that study.

Reviewer 1

The use of IHC to quantitate CD8 T-cells would be clarified simply by including the name of the system used, e.g. HALO. Examples of IHC images would also assist the reader in understanding why they were undertaken, and what was being analyzed. Something along the lines of “In a subset of cases, quantitative image analyses of the density of CD8+ cells were compared to data derived from the Methy|CIBERSORT cell estimates from the same tissue blocks.”

Author response

For the Oslo samples that were stained for CD8 and for which digital quantification of staining was presented, an in-house platform developed in Matlab and described in Salberg UB et al BJC 2022 (DOI: 10.1038/s41416-022-01782-x) was used. We have added this brief description and citation to the Methods section (lines 792-794) and have included a Supplementary Methods document, in which we show example images along with detailed explanation. We hope this clarifies this point.

Comment by reviewer 1b: Thank you for the clarification and for inclusion of the methods.

Reviewer 1

The statement, “The presence of tumor-reactive T-lymphocytes in the tumors and blood of cervical cancer patients are established prognostic factors, with the importance of reactivity against specific HPV peptides demonstrated for both cytotoxic and effector compartments, particularly in deeply-invasive tumors” overstates the findings reported in references 33-41. It does not appear that any of the papers reports tumor-reactive T-lymphocytes in tumors.

Author response

Thank you for bringing this to our attention. We have modified the sentence to more accurately reflect the conclusions of the cited studies as follows (lines 354-356):

“The presence of circulating HPV-reactive T-lymphocytes and of tumour-infiltrating cytotoxic T lymphocytes have been associated with lower N-stage and improved prognosis in cervical cancer patients...”

Comment by reviewer 1b: Ok.

Reviewer 1

Again referring to Table 1, survival by site is not so useful as survival by stage at each site. Survival by stage and C1/C2 would help to clarify the impact of this distinction.

Author response

We have expanded Table S7, which is now entitled “Breakdown of patient numbers, vital status and survival for C1 and C2 patients by tumour stage”. Where possible we have given 5-year survival rates (%) and have noted that for TCGA, this is OS and for the European cohort, this is DSS. Note that in all cases, except for the 14 TCGA patients with stage IV disease, survival rates by stage are higher for C1 than C2. This is discussed in the revised manuscript (lines 223-225).

Comment by reviewer 1b: Thank you for providing the by-stage analyses. Even without robust statistics, the data nevertheless support the fact that the findings are not driven by different stage overrepresentation in C1 vs. C2.

Reviewer 1

Finally, in the end, what insights does this work provide that would inform future investigations, or could lead to changes in treatment strategies?

Author response

Key findings that will inform future investigations:

- 1) The identification of a subset of cervical SCCs (C2) which (although confirmed as SCCs by two independent pathologists in our team, see Table S12), display gene expression profiles, a prevalence of alpha-7 HPV types and genomic alterations that are characteristic of adenocarcinoma. This suggests a possible difference in cell-of-origin and/or transition event (see point 2), in this SCC subset that will be of considerable interest for further investigation (for example by using lineage tracing in mouse models as we suggest in the manuscript).
- 2) The identification of STK11 loss and/or mutation as a frequent event in C2 tumours that may (as has been observed in lung cancer), drive a transition from adenocarcinoma to SCC.
- 3) The identification of potential targets for targeted therapies in this poor prognosis subgroup, including YAP1 and the immune checkpoint proteins NT5E, B7-H3 and PD-L2.
- 4) The identification of 26 significantly mutated genes not previously implicated in cervical SCC will inform functional studies on these genes and their role in cervical cancer pathogenesis, potentially enabling identification of further potential therapeutic targets.
- 5) Although larger numbers are needed for robust within-stage comparisons of C1 and C2 tumours, we observe a clear trend in the survival rates between C1 and C2 by stage (data now added to Table S7). Taking molecular (C1/C2) subtyping into account may therefore allow for more accurate prognostication than current staging and potentially (clearly dependent upon prospective studies) different clinical management of patients with C1 versus C2 tumours.

We feel we have touched upon all the above points in the manuscript without overstating the significance of our findings but we are happy to modify the Discussion and Abstract further if the reviewer and/or editors deem it necessary.

Comment by reviewer 1b: The points above are valid, but are not well-elaborated in the discussion and I think the investigators could provide additional clinical implications of the results. For example, aside from the prognostic significance, C1 vs. C2 distinction may identify patient populations at risk for relapse that require further adjuvant therapy after completion of upfront therapy. There are also implications for exploring C1 vs. C2 as a predictor of response within the context of approved agents (e.g. pembrolizumab), but also within the context of ongoing clinical trials with other agents.

We thank the reviewer for recognising these implications of our work and for suggesting further important points for discussion from the clinical perspective. We have made the following additions to the discussion in an attempt to highlight these and we hope they won't mind that we have paraphrased from their comment in places.

Line 531 onwards:

“Although larger numbers are needed for robust within-stage comparisons of C1 and C2 tumours, we observe a clear trend in the survival rates between C1 and C2 by stage (Table S7). Taking molecular (C1/C2) subtyping into account may therefore allow for more accurate prognostication than current staging and potentially (and clearly dependent upon prospective studies) different clinical management of patients with C1 versus C2 tumours. This could include the identification of patients at risk of relapse and who may therefore require further adjuvant therapy after completion of upfront therapy.”

Line 590 onwards:

“Finally, we hope the identification of 21 SMGs that have not previously been implicated in CSCC will stimulate functional studies on these genes and their role in cervical cancer pathogenesis,

potentially enabling identification of new therapeutic targets. The identification of C2-specific alterations to YAP1 and upstream Hippo signalling pathway components is of particular interest, given recent studies that highlight the importance of this pathway in cervical carcinogenesis [<https://doi.org/10.15252/emmm.201404976>; <https://doi.org/10.1016/j.celrep.2019.02.004>; <https://doi.org/10.1038/s41388-022-02390-y>; <https://doi.org/10.7554/eLife.75466>].

Line 618 onwards:

“It may therefore be informative to explore use of the C1 / C2 classification as a predictor of response to pembrolizumab, and within the context of ongoing clinical trials with other agents.”

Final discussion sentence (new text underlined here):

“This suggests that the findings and underlying principle: that CSCC can develop along two trajectories associated with differing clinical behaviour that can be identified using defined gene expression or DNA methylation signatures, are of broad relevance and that they may guide improved clinical management of cervical cancer patients.”

For points (1) and (2) that we’ve made above, regarding *STK11* mutation and the potential transition from adenocarcinoma to SCC during the development of C2 tumours, we have covered this in detail in lines 534-565.